# An electrostatic switching mechanism to control the lipid transfer activity of Osh6p

Nicolas-Frédéric Lipp [1], Romain Gautier [1], Maud Magdeleine[1], Maxime Renard[1], Véronique Albanèse [2], Alenka Čopič [2] & Guillaume Drin [1]

A central assumption is that lipid transfer proteins (LTPs) bind transiently to organelle membranes to distribute lipids in the eukaryotic cell. Osh6p and Osh7p are yeast LTPs that transfer phosphatidylserine (PS) from the endoplasmic reticulum (ER) to the plasma membrane (PM) via PS/phosphatidylinositol-4-phosphate (PI4P) exchange cycles. It is unknown how, at each cycle, they escape from the electrostatic attraction of the PM, highly anionic, to return to the ER. Using cellular and in vitro approaches, we show that Osh6p reduces its avidity for anionic membranes once it captures PS or PI4P, due to a molecular lid closing its lipid-binding pocket. Thus, Osh6p maintains its transport activity between ER- and PM-like membranes. Further investigations reveal that the lid governs the membrane docking and activity of Osh6p because it is anionic. Our study unveils how an LTP self-limits its residency time on membranes, via an electrostatic switching mechanism, to transfer lipids efficiently.

[1] Université Côte d'Azur, CNRS, Institut de Pharmacologie Moléculaire et Cellulaire, 660 route des lucioles, 06560 Valbonne, France. [2] Institut Jacques Monod, CNRS, Université Paris Diderot, Sorbonne Paris Cité, 75205 Paris, France. Correspondence and requests for materials should be addressed to G.D. (email: drin@ipmc.cnrs.fr)

Lipids are accurately distributed in the eukaryotic cell to confer to organelles their physical features and molecular identity[1,2]. Phosphatidylserine (PS), representing 2–10% of total membrane lipids[3–6], is a negatively-charged glycerophospholipid that is spread along a gradient in the cell. Its concentration increases between the endoplasmic reticulum (ER), where it accounts for 5–7% of glycerophospholipids, and the plasma membrane (PM) where its proportion can rise up to 30%[7–9]. It mostly accumulates in the cytosolic leaflet of this membrane. This build-up and asymmetric distribution of PS in the PM are critical for signaling pathways, mediated by cytosolic proteins that are recruited and/or activated by this lipid[9]. Like other lipids predominantly found in the PM[10–13], PS is synthesized in the ER[14], meaning that it must be exported to the PM. How this is accomplished was unknown[9] until the demonstration that in *Saccharomyces cerevisiae*, Osh6p and its homolog Osh7p mediate ER-to-PM PS transport[15].

Osh6p and Osh7p belong to the oxysterol-binding protein-related proteins (ORPs)/oxysterol-binding homologues (Osh) family which is a major group of lipid transfer proteins (LTPs) and/or lipid sensors in eukaryotes[16]. All members of this family have a domain—called ORD (OxySterol Binding Protein (OSBP)-Related Domain)—to encapsulate specific lipid species. A few of them, including Osh6p and Osh7p, consist only of an ORD[15,17], whereas others contain additional modules[16]. The ORD of Osh6p/Osh7p has a cavity to host one PS molecule; this pocket is closed by an N-terminal lid that shields the lipid from the aqueous environment[15]. Remarkably, Osh6p can alternately trap phosphatidylinositol-4-phosphate (PI4P)[18]. PI4P is made at the PM, as well as in other compartments of the secretory pathway, and is hydrolysed into phosphatidylinositol at the ER[19]. Thus a PI4P gradient exists at the ER/PM interface and Osh6p/Osh7p use this gradient to transfer PS from the ER to the PM by PS/PI4P exchange cycles. During each cycle, they would extract a PS molecule from the ER, exchange it for PI4P at the PM, and deliver PI4P to the ER where PI4P is hydrolyzed[18]. In human, ORP5/ORP8 act similarly at ER-PM contact sites[20]. Several ORP/Osh proteins likely use PI4P gradients to move other lipids, as demonstrated for Osh4p and OSBP, which are sterol/PI4P exchangers[21].

Due to the abundance of PS, the cytosolic side of the PM is highly negatively-charged compared to that of other organelles[9,22,23]. This generates high electrostatic forces that are exploited by signaling proteins targeting this region: MARCKS[24], Src[25], K-Ras[26] or Rac1[27] bind to the PM via a stretch of positively-charged amino-acids and a lipidic tail; conventional PKC[28] recognizes PS using a C2 domain. These binding reactions are regulated: MARCK or K-Ras are displaced from the PM through phosphorylation[26,29], whereas the binding of the PKC C2 domain depends on $Ca^{2+}$ levels[28]. What about Osh6p and Osh7p? During one PS/PI4P exchange cycle, they must interact with the PM and then escape from the electrostatic field of the PM to return to the ER, whose surface is less anionic[1,8,9,22]. It is often implicitly assumed that LTPs must bind transiently to donor and acceptor organelle membranes to be efficient[30–33]. Because structural analyses suggest that several LTPs change their conformation when encapsulating a ligand[17,34,35], it is also often speculated that the status of an LTP, empty or loaded, influences its membrane-binding capacities[17,33]. This has been demonstrated for the phosphatidylinositol transfer protein PITPα[36] whose exchange dynamics depend on its ability to undergo a conformational change upon capturing lipid. Apart from this example, the dynamic changes during transport reactions are poorly documented. Furthermore, there are many examples of LTPs (Sec14p, nsLTP, GLTP) whose activity is impeded in the presence of negatively-charged membranes due to their high

retention on such membranes[37–39]. How can Osh6p/Osh7p work efficiently at the ER/PM interface?

Here, we address this issue via cellular observations combined with in vitro and in silico analyses. We find that Osh6p has a reduced binding to anionic membranes, once it extracts PS or PI4P, because its N-terminal lid closes. Consequently, Osh6p maintains a fast transport activity between membranes resembling the ER-membrane and PM. Molecular dynamics (MD) simulations suggest that the attraction forces between the closed form of Osh6p and membranes are weak, because the lid is anionic and masks some membrane-interacting regions. Charge-neutralizing mutations in a aspartate(D)/glutamate(E)-rich motif in the lid impair Osh6p's activity. We demonstrate that a LTP uses an electrostatic switching mechanism to self-limit its residency time on membranes and thereby, be efficient.

## Results

**Osh6p firmly binds to membranes if its lid region is mutated.** When Osh6p is chromosomally tagged with GFP, it can be observed in patches at the cell cortex, confirming its enrichment at ER-PM contact sites[15,40]. In addition, diffuse fluorescence throughout the cell suggests a significant soluble pool of Osh6-GFP (Fig. 1a). Similar localization is observed when Osh6p-GFP is expressed from a CEN plasmid under the control of the pADH1 promoter in an *osh6Δosh7Δ* strain in the case of weakly expressing cells (Fig. 1b). In contrast, cells with higher fluorescent levels, likely harboring multiple copies of the plasmid, display only cytoplasmic fluorescence, suggesting that there are a limited number of Osh6p-binding sites at the cortical ER. We next compared localization of Osh6p (termed WT), fused to mCherry, to that of two mutants, L69D and H157A/H158A (thereafter called HH/AA), which are deficient for PS transport[18]. Osh6p (L69D), in which an anionic residue replaces a hydrophobic one in the lid (Fig. 1c), cannot bind or trap PS[15]. Osh6p(HH/AA) does not recognize PI4P and is therefore unable to convey PS via PS/PI4P exchange[18]. Despite their similar defects in PS transport, these mutants have different intracellular distribution (Fig. 1d): Osh6p(HH/AA) is cytosolic and enriched at the cortex like Osh6p, whereas Osh6p(L69D) is less cytosolic and instead bound to membranes in highly-expressing cells. Notably, it colocalizes with the ER-marker Sec63p, but is also seen in other intracellular structures and enriched at the cell surface (Fig. 1d and Supplementary Fig. 1a). Besides, some bright internal spots may represent Osh6p(L69D) aggregates (Fig. 1d). The localization of a deletion mutant Osh6pΔ69 that lacks the lid (residues 36 to 69) and an upstream low-complexity region [1–35], is similar to that of Osh6p(L69D) (Fig. 1d and Supplementary Fig. 1a). This mutant does not transfer PS in vitro[18] or in cells (Supplementary Fig. 1b). Thus, a mutation within the lid, or the deletion of the N-terminal region that includes the lid, increases the propensity of Osh6p to interact with membranes.

We next analyzed the membrane-binding capacity of Osh6p in vitro by measuring its association with neutral or anionic liposomes in flotation assays. Recombinant wild-type protein was mixed with liposomes made of 1,2-dioleoyl-*sn*-glycero-3-phosphocholine (DOPC) and increasing levels (0, 5, 10 or 30% mol/mol) of 1-palmitoyl-2-oleoyl-*sn*-glycero-3-phosphoserine (POPS). These liposomes were then recovered from the top of a sucrose gradient following centrifugation, and the percentage of Osh6p bound to liposomes was quantified. Osh6p has a low avidity for membranes; <15% of protein was recruited onto liposomes irrespective of their PS density (Fig. 1e). We then evaluated how Δ69, L69D and HH/AA mutants interacted with membranes containing 30% PS, or additionally 4% PI4P, compared to Osh6p(WT). Remarkably, Δ69 and L69D mutants

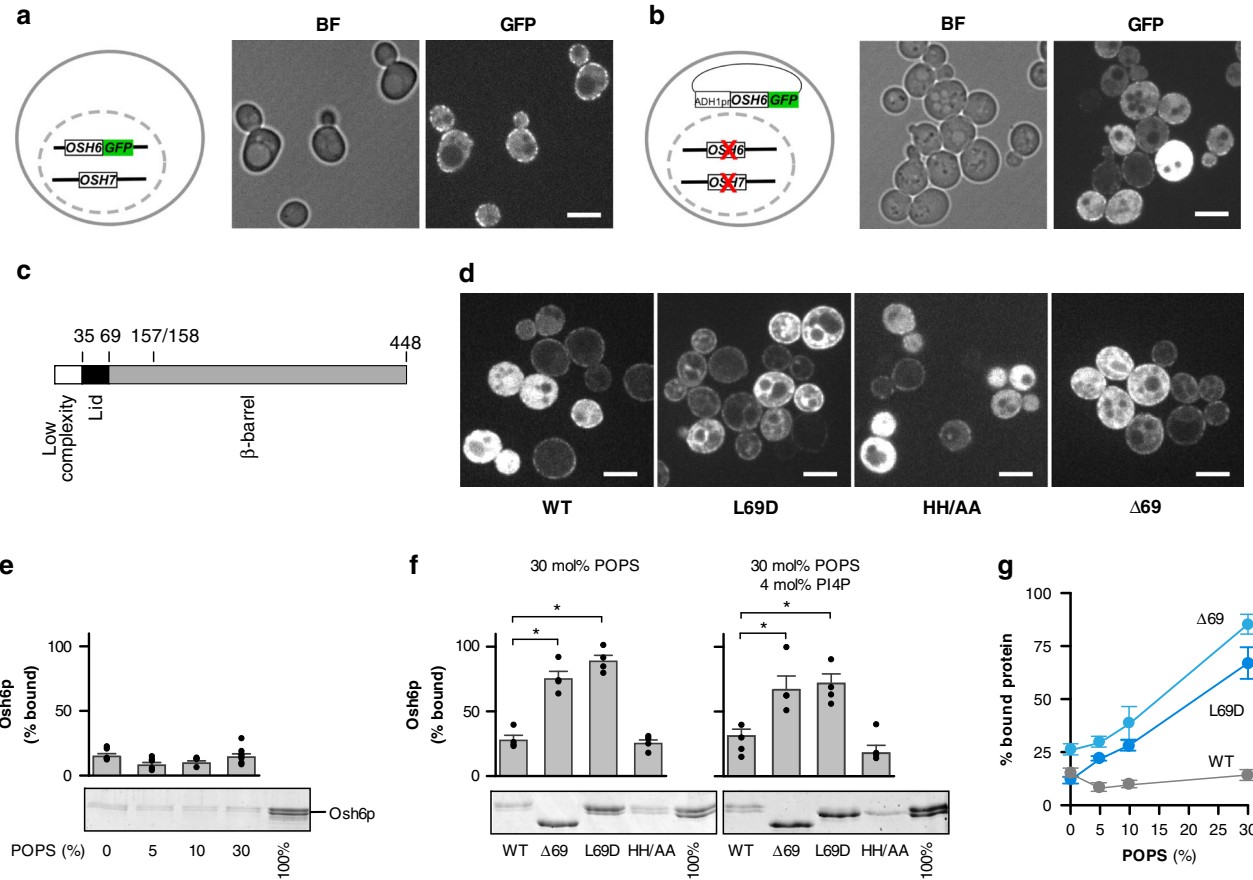

**Fig. 1** Osh6p weakly binds to membranes due to its lid region. **a** Localization of chromosomally-tagged Osh6p-GFP in wild-type yeast, as depicted in the scheme. Brightfield (BF) and GFP fluorescence images are shown. Scale bar: 5 μm. **b** Localization of Osh6p-GFP, under the control of an pADH1 promoter expressed from a low-copy plasmid in osh6Δosh7Δ cells, as depicted in the scheme. Level of Osh6-GFP expression varies due to variability in plasmid copy number per cell. Scale bar: 5 μm. **c** Representation of the different sub-regions of Osh6p showing the position of mutations and deletions in Osh6p mutants. **d** Localization of Osh6p WT and mutants fused to mCherry expressed from a plasmid in osh6Δosh7Δ cells as in (**b**). Scale bar: 5 μm. **e** Flotation assay. Osh6p (750 nM) was incubated in HK buffer at 25 °C for 10 min with DOPC liposomes (750 μM lipids) containing 0, 5, 10 or 30% POPS at the expense of DOPC. After centrifugation, the liposomes were recovered at the top of a sucrose cushion and analyzed by SDS-PAGE. The amount of protein recovered in the top fraction (lane 1 to 4) was quantified and the fraction of liposome-bound Osh6p was determined using the content of lane 5 (100%) as a reference. Experiments were repeated (n = 4–7) using different batches of extruded liposomes. **f** Osh6p, Osh6pΔ69, Osh6p(L69D) or Osh6p(HH/AA) was added to DOPC liposomes doped with 30% POPS or to DOPC/POPS/diC16:0-PI4P (66/30/4 mol/mol) liposomes (n = 4, *P < 0.05, Mann–Whitney test). **g** Percentage of membrane-bound Osh6p, Osh6pΔ69 or Osh6p(L69D) as a function of the density of POPS in liposomes (n = 4–7 for Osh6p WT, n = 5–6 for Osh6pΔ69, n = 5–7 for Osh6p(L69D)). Error bars correspond to s.e.m. Source data are provided as a Source Data file

have a high avidity for these membranes (~75% of bound protein, Fig. 1f) whereas Osh6p(HH/AA) remains mostly soluble, like Osh6p. The fraction of membrane-bound Osh6pΔ69 and Osh6p(L69D) increases with the density of PS in the liposomes, contrary to Osh6p (Fig. 1g), suggesting that electrostatic forces drive the interaction of these mutants with membranes. We conclude that the increased localization of L69D and Δ69 mutants to cellular membranes might be due to their stronger avidity for lipid bilayers.

**High membrane-binding Osh6p's mutants do not extract lipids**. To explain these data, we looked for functional differences between Osh6p and its mutants, first by quantifying their ability to remove PS from liposomes using a PS-sensor NBD-C2$_{Lact}$[18]. This sensor bears an NBD fluorophore whose emission intensity depends on whether it is bound or not to membrane. When incubated with liposomes containing 2% POPS, the fluorescence of NBD-C2$_{Lact}$ was high (Fig. 2a), indicating that it was membrane-bound. In the presence of Osh6p, the fluorescence was low and comparable to that measured with PS-free liposomes.

This indicated, after signal normalization, that 75–80% of accessible PS was extracted (Fig. 2b). Osh6p(HH/AA) exhibited a similar extraction efficiency. In contrast, Osh6pΔ69 and Osh6p (L69D) did not provoke any substantial dissociation of the probe, indicative of their inability to extract PS. A related assay based on a PI4P-sensor, NBD-PH$_{FAPP}$[41], revealed that Δ69, L69D and HH/AA mutants fail to extract PI4P (Fig. 2c, d, <20% extraction), contrary to Osh6p (100%). Jointly these data indicated that Osh6pΔ69 and Osh6p(L69D) extract neither PS nor PI4P, whereas Osh6p(HH/AA) is still able to encapsulate PS. We conclude that the higher avidity of Osh6pΔ69 and Osh6p(L69D) for membranes, in yeast or in vitro, might be because they remain empty, although these membranes contain PS and PI4P.

**Osh6p strongly binds to anionic membranes as its lid is open**. Osh6pΔ69 remains empty probably because, without a lid, it cannot secure PS or PI4P inside its pocket, as suggested by previous structural and in silico analyses[15,18]. For the L69D mutant, we assumed that it behaved like Osh6pΔ69 because its lid was unable to close. To investigate this point, we devised an assay for

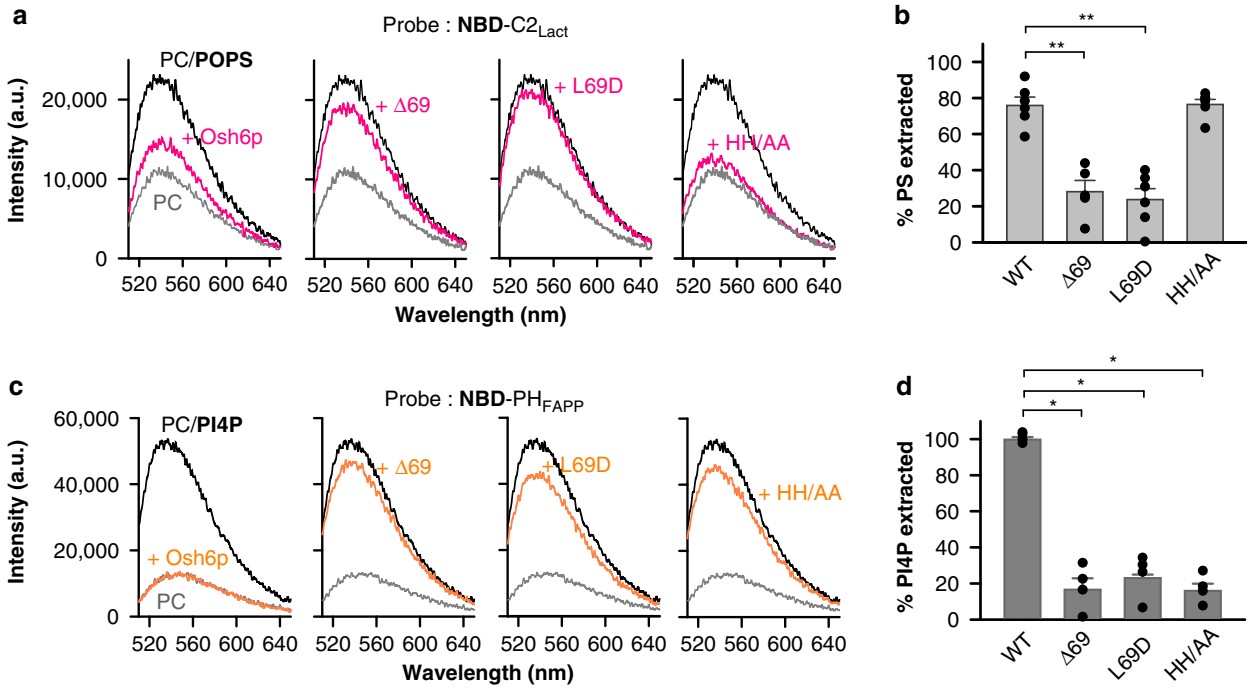

**Fig. 2** Osh6p mutants extract differently PS and PI4P. **a** PS extraction assay. Fluorescence spectra of NBD-C2$_{Lact}$ (250 nM) measured upon excitation at 460 nm in the presence of liposomes (80 µM, 2% POPS) before (black spectrum) and after (pink spectrum) adding Osh6p or mutants (3 µM). A reference spectrum (in gray) was recorded with pure DOPC liposomes incubated with NBD-C2$_{Lact}$. **b** Percentage of accessible PS extracted by each construct based on signal normalization at 536 nm. ($n = 5$–6, $**P < 0.01$, Mann–Whitney test). **c** PI4P extraction assay. Fluorescence spectra of NBD-PH$_{FAPP}$ (250 nM) ($\lambda_{ex} = 460$ nm) mixed with liposomes (80 µM, 2% diC16:0-PI4P) before (black spectrum) and after (orange spectrum) adding 3 µM Osh6p WT or mutants. A reference spectrum of NBD-PH$_{FAPP}$ (in grey) was recorded with PI4P-free liposomes. **d** Percentage of accessible PI4P extracted by each construct based on signal normalization at 536 nm ($n = 4$, $*P < 0.05$, Mann–Whitney test). Error bars correspond to s.e.m. Source data are provided as a Source Data file

evaluating the conformational state of Osh6p. We purified an Osh6p(noC/S190C) construct including a single cysteine at position 190, predicted to be solvent-exposed only if the lid is open (Fig. 3a). This protein was mixed with 7-diethylamino-3-(4′-maleimidylphenyl)-4-methylcoumarin (CPM), which becomes fluorescent when forming a covalent bond with a thiol group. After a 30-min incubation with the protein alone in buffer, we measured a high fluorescence indicative of the accessibility of the cysteine. This suggested that Osh6p could regularly adopt an open state (Fig. 3a, Supplementary Fig. 2a, b). A higher signal was obtained when Osh6p(noC/S190C) was mixed with pure DOPC liposomes, indicating that the protein was more frequently open in that context (Fig. 3a, Supplementary Fig. 2b). In contrast, when Osh6p(noC/S190C) was added to liposomes with 2% POPS or PI4P, we recorded an extremely low signal, suggesting that the protein, in the presence of its ligands, remained essentially closed.

We repeated these measurements with an Osh6p(noC/S190C) construct bearing the L69D mutation. We measured in solution a CPM signal similar to that obtained with the previous construct and obtained higher signals with liposomes, but even if they contained PS or PI4P. This was not due to a denaturation process since the two noC/S190C constructs are properly folded (Supplementary Fig. 2c). We conclude that the L-to-D substitution prevents the lid from closing in the presence of a ligand and that the L69D mutant binds more to PS and PI4P-containing membranes because it corresponds to a constitutively open form of Osh6p.

Therefore, we hypothesized that the affinity of Osh6p for membranes was dependant on its conformational state. To investigate this, we performed assays with Osh6p and liposomes incorporating diphytanoyl-PS instead of POPS. Diphytanoyl-PS is a synthetic PS species whose C16:0 acyl-chains bear branched methyl groups (Fig. 3b). Consequently, it would be too bulky to

be accommodated by Osh6p. Our PS-extraction assay confirmed this, showing no change in the signal of NBD-C2$_{Lact}$ when Osh6p was mixed with liposomes containing 2% diphytanoyl-PS (Fig. 3c). We mixed Osh6p(noC/S190C) with these liposomes and found using our CPM assay that the protein was unable to properly close because the final CPM intensity was similar to that measured with DOPC liposomes and much higher than in the presence of membranes with POPS (Fig. 3d). In flotation assays, Osh6p weakly bound to neutral membranes exclusively made of DOPC or containing 30% diphytanoyl-PC, or to anionic membranes with 30% POPS (<25% of membrane-bound protein). In contrast, it tightly associated with anionic liposomes with 30% diphytanoyl-PS ($72.6 \pm 3.1\%$). We confirmed this using a second assay and a fluorescent Osh6p in which a unique cysteine (C262) within the β14-β15 loop, is labeled with NBD (Supplementary Fig. 3a–c). Adding DOPC/diphytanoyl-PS (70/30 mol/mol) liposomes to NBD-Osh6p elicited a blue-shift of its fluorescence with an increase in intensity, indicative of its recruitment onto membrane (Fig. 3f). In contrast, no change was seen with liposomes solely made of DOPC or enriched with 30% diphytanoyl-PC or POPS. Last, we found that Osh6p strongly bound to liposomes enriched with other anionic lipids, either 30% phosphatidic acid (PA) or phosphatidylinositol (PI) (Supplementary Fig. 4a). They are not ligands for Osh6p as, unlike PS, they do not prevent the protein from extracting PI4P (Supplementary Fig. 4b) when they are in excess in the same membrane. We conclude that Osh6p binds in a non-specific electrostatic manner to anionic membranes, if they are devoid of extractable lipid, because it remains empty and open.

**Osh6p dissociates from membranes upon trapping its ligands.** Our data suggested that Osh6p loses its ability to associate with a

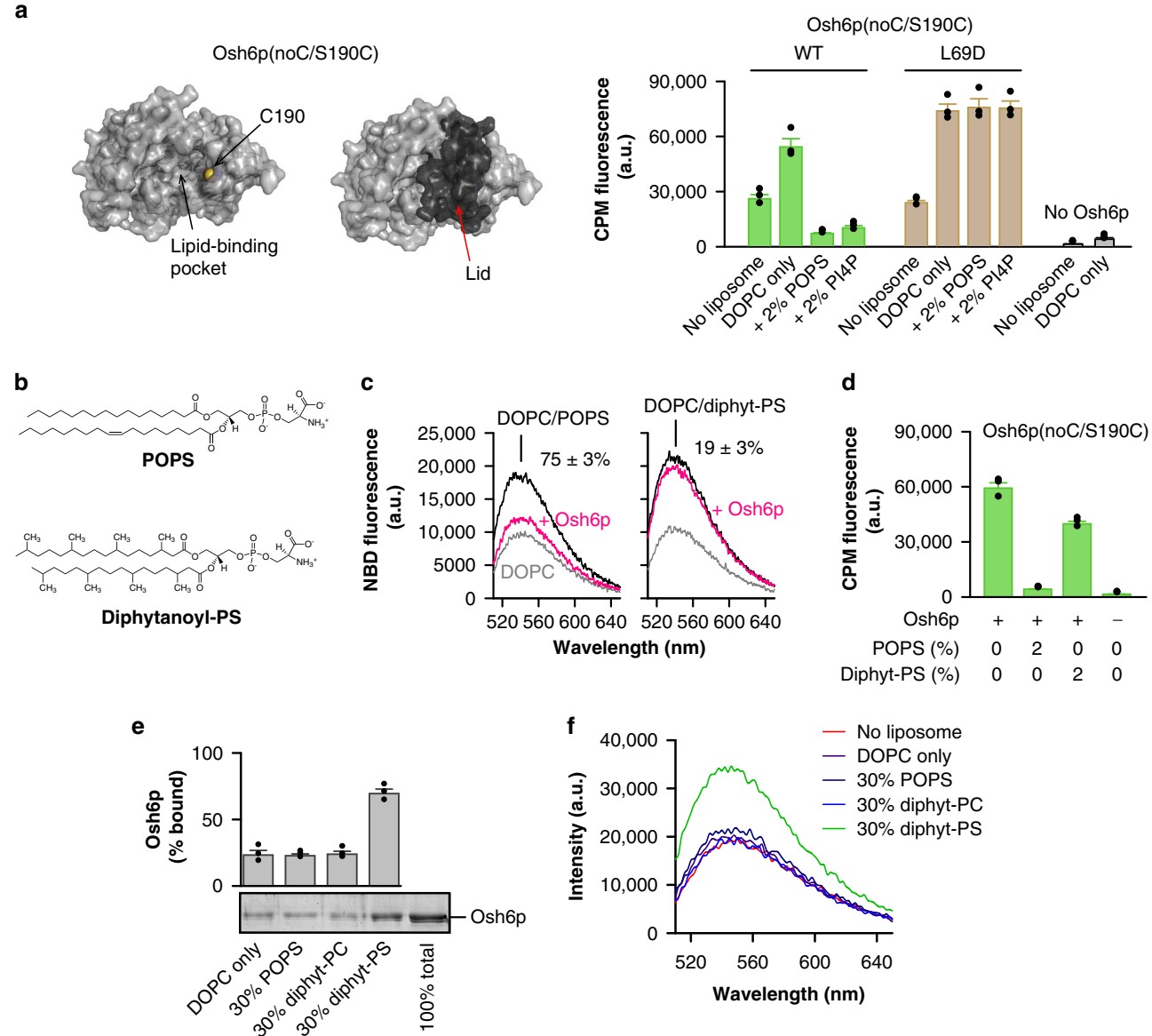

**Fig. 3** The binding of Osh6p to anionic membranes depends on its conformational state. **a** Accessibility CPM assay. Osh6p(noC/S190C) has a unique cysteine which is solvent-accessible only if the lid is open. Osh6p(noC/S190C) or its L69D counterpart (400 nM) was incubated in HK buffer at 30 °C with DOPC liposomes (400 μM lipids) including or not 2% POPS or diC16:0-PI4P. CPM (4 μM) was added and the apparition of fluorescence was measured over time. Bars correspond to the fluorescence (measured at $\lambda = 470$ nm) 30 min after adding CPM ($n = 3$). **b** Chemical structure of POPS and diphytanoyl-PS. **c** PS extraction assay. Fluorescence spectra of NBD-C2$_{Lact}$ (250 nM) ($\lambda_{ex} = 460$ nm), mixed with DOPC liposomes (80 μM lipid) incorporating 2% POPS or 2% diphytanoyl-PS, were recorded before and after adding Osh6p (3 μM). A control experiment was done with pure DOPC liposomes. Normalized signal provides the fraction (in percentage) of extracted PS ($n = 4$ with POPS, $n = 3$ with diphytanoyl-PS). **d** Accessibility assay. Osh6p(noC/S190C) (400 nM) was incubated with DOPC liposomes doped or not with 2% POPS or diphytanoyl-PS and mixed with CPM (4 μM). Intensity bars correspond to the fluorescence measured 30 min after adding CPM ($n = 3$). **e** Flotation assay. Osh6p (750 nM) was incubated with pure DOPC liposomes (750 μM lipids) or liposomes containing 30% POPS, diphytanoyl-PC or diphytanoyl-PS at the expense of DOPC. The bars represent the fraction of protein bound to each type of liposomes ($n = 3-4$). **f** Emission spectra of NBD-labeled Osh6p (200 nM, excitation at 460 nm) in HK buffer measured in the absence (red spectrum) or presence of liposomes (400 μM total lipids) made only of DOPC or including 30% POPS, diphytanoyl-PC or diphytanoyl-PS (at the expense of DOPC). The contribution of buffer alone or of light scattering from liposomes to the fluorescence signal was subtracted. Error bars correspond to s.e.m. Source data are provided as a Source Data file

membrane once it extracts a lipid ligand and closes. To explore this, we first examined whether the lid was the sole structural element of the N-terminus of Osh6p to mediate this process. We purified a Δ35 deletion mutant, devoid of the low-complexity region upstream of the lid, and found that it behaved similarly as Osh6p. First, it efficiently extracts PS or PI4P, except if it bears the L69D mutation (Fig. 4a). Second, compared to Δ69 and L69D mutants, it weakly interacts with anionic membranes

containing PS or both PS and PI4P (Fig. 4b, Supplementary Fig. 4c). Third, fluorescence assays showed that an Osh6pΔ35 construct, labeled with NBD, binds to diphytanoyl-PS-rich membranes but not to neutral or POPS-rich membranes (Fig. 4c). This suggests that in the N-terminal region of Osh6p, the lid is the unique element that guarantees the extraction efficiency of the protein and controls its membrane-binding properties.

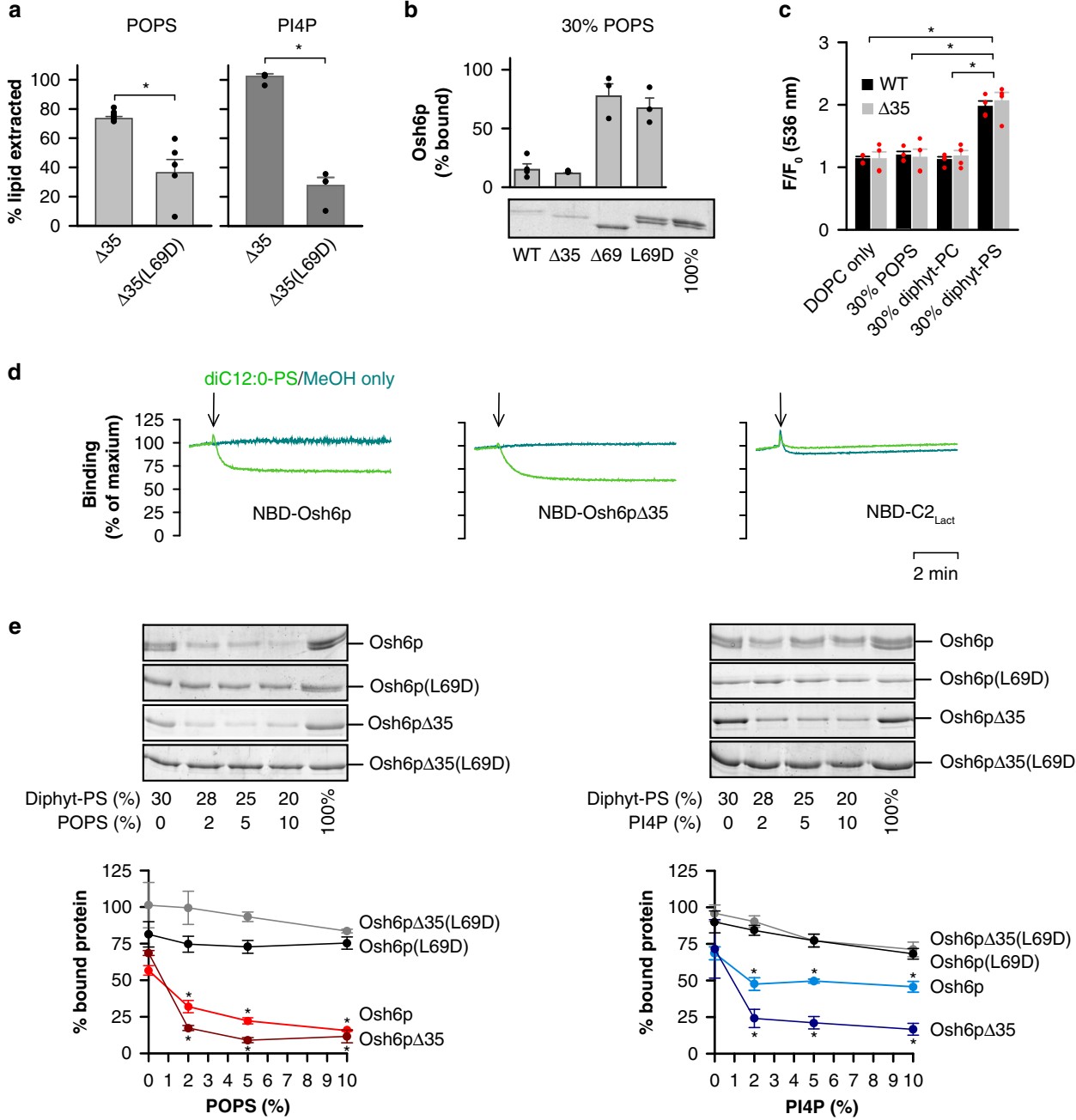

**Fig. 4** Osh6p dissociates from anionic membranes when it extracts a lipid ligand. **a** PS and PI4P extraction assay. Percentage of accessible PS or PI4P extracted by 3 µM Osh6pΔ35 or Osh6pΔ35(L69D) from liposomes (80 µM lipids, with 2% POPS or diC16:0-PI4P) using NBD-C2$_{Lact}$ or NBD-PH$_{FAPP}$, respectively. Values are obtained by normalizing the signal at 536 nm ($n = 5$–6 for PS extraction assays, $n = 4$ for PI4P extraction assays, **$P < 0.01$, *$P < 0.05$, Mann–Whitney test). **b** Flotation assay. Osh6p, Osh6pΔ35, Osh6pΔ69 or Osh6p(L69D) (750 nM) was added to liposomes (750 µM lipids) with 70 mol% DOPC and 30 mol% POPS ($n = 3$–4). **c** Binding ability of NBD-labeled wild-type Osh6p (black bar) or Δ35 mutant (grey bar) to liposomes, as function of their composition, was given by the ratio of the NBD fluorescence intensity, F and $F_0$, measured at 536 nm in the presence or absence of liposomes, respectively ($n = 4$, *$P < 0.05$, Mann–Whitney test). **d** Dissociation of NBD-labeled Osh6p WT or Δ35 from anionic liposomes (DOPC/diphytanoyl-PS 7/3 mol/mol) upon adding extractable PS. Each protein (100 nM) was pre-mixed with liposomes (400 µM lipids) in HK buffer at 30 °C. Then, a small volume (1 µL) of a stock solution of diC12:0-PS in methanol or of pure methanol was injected. NBD fluorescence was followed at 535 nm ($\lambda_{ex}$ = 460 nm) over 10 min and normalized. Each trace is representative of several experiments. Control measurements were performed with NBD-C2$_{Lact}$ (250 nM) instead of NBD-labeled Osh6p. **e** Flotation assay. Osh6p, Osh6pΔ35, Osh6p(L69D) or Osh6pΔ35(L69D) at 750 nM was incubated with DOPC/diphytanoyl-PS (7/3 mol/mol) liposomes (750 µM lipids) containing increasing amount of POPS (0, 2, 5 or 10%). Alternatively, each protein was incubated with DOPC/diphytanoyl-PS (7/3 mol/mol) membranes with 0, 2, 5 or 10% diC16:0-PI4P ($n = 4$, *$P < 0.05$, Mann–Whitney test, comparison with the data measured with no PS or PI4P). Error bars correspond to s.e.m. Source data are provided as a Source Data file

Then, we examined in real-time whether the membrane-binding capacity of Osh6p changed in the presence of a ligand. Osh6p or the Δ35 mutant, labeled with NBD, was mixed with liposomes containing 30% diphytanoyl-PS, resulting in a maximal NBD signal, indicative of their association to membranes. The addition of short-chained PS (diC12:0) stored in methanol provoked a fast decrease in fluorescence, whereas pure methanol did not (Fig. 4d). This effect was specific to Osh6p since no drop in signal was seen when adding PS to NBD-C2$_{Lact}$ bound to diphytanoyl-PS-rich liposomes. Thus, Osh6p dissociates from an anionic membrane once an extractable lipid is available.

To quantify this, we performed flotation assays by incubating Osh6p or Osh6pΔ35 with liposomes containing 70% DOPC and 30% diphytanoyl-PS, and no or increasing amount of POPS (2, 5 or 10%, Fig. 4e). Remarkably, whereas these proteins efficiently bind to membranes devoid of POPS, they bind two-time less to membranes doped with 2% POPS and are essentially soluble at higher POPS concentration (Fig. 4e). Comparable results were obtained with diphytanoyl-PS-rich liposomes doped with PI4P, yet at a lesser extent with Osh6p. In contrast, Osh6p(L69D) and Osh6pΔ35(L69D) remained attached, whatever the POPS or PI4P level. Last, Osh6p and Osh6pΔ35 associate with anionic liposomes rich in PA or PI but, overall, less if these liposomes contain 5% PS or PI4P (Supplementary Fig. 4d). Jointly, these data suggest that trace amounts of ligand reduce the retention of Osh6p on anionic membranes, if its lid is functional.

**Osh6p retains its solubility and activity as it can close**. We further explored whether the low affinity of the lipid-loaded form of Osh6p for membranes might be important for its exchange activity at the ER/PM interface. To this end, we tested the ability of Osh6p, Osh6pΔ35 or Osh6pΔ69 to transport PI4P from L$_B$ (PM-like) liposomes doped with 5% PI4P and Rhodamine-phosphatidylethanolamine (Rhod-PE), to L$_A$ liposomes (ER-like) containing 5% PS, under conditions where PS/PI4P exchange occur[18]. Two conditions were tested for each protein: with L$_B$ liposomes incorporating 25% diphytanoyl-PC or 25% diphytanoyl-PS, thus either weakly or highly anionic. In each measurement, the PI4P-sensor NBD-PH$_{FAPP}$[41] was pre-mixed with L$_B$ liposomes: it associates with these vesicles resulting in the quenching of the NBD signal by FRET to Rhod-PE. Then, L$_A$ liposomes and an Osh6p variant were injected (Fig. 5a). With weakly anionic L$_B$ liposomes, the addition of Osh6p or Osh6pΔ35 provoked a fast fluorescence dequenching, corresponding to the transfer of PI4P from L$_B$ to L$_A$ membranes. According to the normalized signals, PI4P is rapidly equilibrated between liposomes. The initial transport rates for Osh6p and its Δ35 counterpart were 31.2 ± 4 and 30.2 ± 1.8 PI4P min$^{-1}$ per protein, respectively (mean ± s.e.m., Fig. 5b). Osh6pΔ69 transports PI4P to some extent, despite the absence of the lid (3.3 ± 1.2 PI4P min$^{-1}$), as found previously[18]. With very anionic L$_B$ liposomes, Osh6p WT and Δ35 remained efficient as PI4P was almost completely transferred within 2 min. The transport rate was reduced by ~70% with Osh6p but only by ~30% with Osh6pΔ35. In comparison, the activity of Osh6pΔ69 was abrogated. In view of our binding measurements, we conclude that Osh6p and Osh6pΔ35, contrary to Osh6pΔ69, might minimize their retention onto anionic L$_B$ liposomes owing to the lid and the presence of PI4P. Consequently, they can efficiently shuttle PI4P to L$_A$ liposomes in the context of PS/PI4P exchange.

We then investigated whether Osh6p or Osh6pΔ35 remained soluble during transport by measuring to what extent their NBD-labeled forms bound to L$_A$ and L$_B$ liposomes, respectively doped with 5% PS and 5% PI4P (as in the transport assays), or not. With L$_B$ liposomes containing 25% diphytanoyl-PC, each protein

remained mostly in solution regardless of the presence or absence of ligand, as we observed only a slight increase in NBD fluorescence above its minimal value in the two cases (Fig. 5c). In comparison, a major difference was seen with L$_B$ liposomes rich in diphytanoyl-PS. If PS and PI4P were present, proteins were weakly membrane-bound, as previously seen. In contrast, if both ligands were absent, proteins were fully bound, as indicated by a high signal, which became noisy due to liposomes aggregation. We conclude that, when lipid ligands are present, Osh6p remains free to circulate between membranes to exchange them.

**The lid is anionic and weakens the membrane binding of Osh6p**. We next examined how the lid, when closing, could change the membrane-binding ability of Osh6p. Looking at the Osh6p structure, which corresponds to Osh6pΔ35[15], we identified a motif, rich in aspartate and glutamate at the N-terminal end of the lid. Analyses of ORP/Osh sequences with a PS-recognizing motif LPTFILE[15] indicated that the D/E-rich stretch is conserved in Osh6p and Osh7p homologs (Supplementary Fig. 5), but also in ORP9, −10 and −11, suggesting a functional role. A comparison of the electrostatic potential surface of Osh6pΔ35, in complex with PS, and that of Osh6pΔ69, in a ligand-free state, revealed that the lid exposes a negative surface (Fig. 6a) and hides a positively-charged area around the entry of the lipid-binding pocket. Thus, the closing of the lid might modulate the avidity of Osh6p for membrane by altering the electrostatic features of its surface.

To explore this possibility, we analyzed by MD simulations how Osh6pΔ35 or Δ69 docked onto an anionic membrane made of 70% DOPC and 30% POPS (four simulations of 500 ns per protein, Supplementary Movie 1 to 4). Each construct was placed, with diverse initial orientations, ~54 Å away from the membrane plane (Supplementary Fig. 6a). Root-Mean-Square-Deviation calculation and secondary structure analysis showed that Osh6p remained folded during all trajectories except at the end of the Δ69–4 trajectory (the C-terminal part of Osh6pΔ69 displaced from the core structure, Supplementary Figs. 6b, 7). In each Osh6p variant, the N and C-termini, several loops and the α7-helix are slightly mobile, as indicated by the Root-Mean-Square Fluctuations per residue; in Osh6pΔ35, the lid displayed some motions (Supplementary Fig. 6c). Key differences were found when analyzing how Osh6pΔ35 and Δ69 bound to the membrane. In all trajectories, Osh6pΔ69 associates with the membrane, as reflected over time by a shorter height (h) of its center of mass relative to the membrane plane (shortest h = 16.1 Å, Fig. 6b). In contrast, Osh6pΔ35 binds to the membrane during two trajectories (Δ35-1 and −4, shortest h = 29 Å) but remains in solution during the two others (Δ35-2 and −3, h ≥ 41.7 Å). Calculating Coulomb and VDW interaction energy between each protein and the membrane revealed a major role of electrostatic attraction in the binding process (Supplementary Fig. 8). Remarkably, the Coulomb and VDW interaction between Osh6pΔ69 and the membrane were more than twice stronger (lowest E = −2,454 and −687 kcal mol$^{-1}$, respectively) compared to Osh6pΔ35 (lowest E = −1,173 and −352 kcal mol$^{-1}$). These results support the idea that Osh6p is more prone to bind to a membrane when its lid is open.

Osh6pΔ35 and Δ69 are differently positioned on the membrane at the end of simulations, as indicated by the angle measured between the α7-helix of the protein and the normal to the membrane plane (Fig. 6b). To better analyze this, we counted during trajectories how many times each of their residues was less than 5 Å away from lipid headgroups, considering there was a contact with membrane below this threshold. Osh6pΔ35 docks

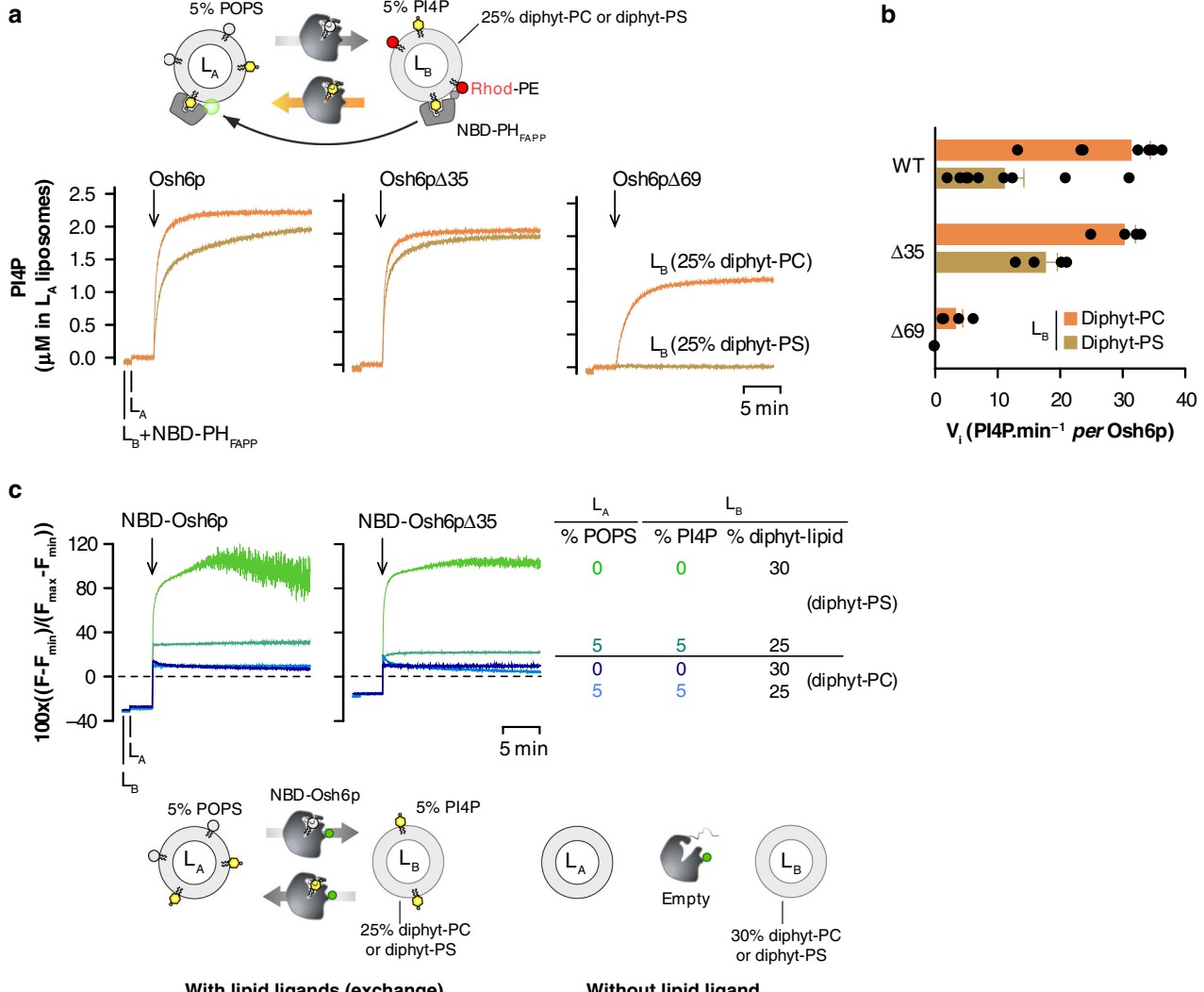

**Fig. 5** Osh6p remains mostly soluble during PS/PI4P exchange between anionic membranes. **a** PI4P transport assay. $L_B$ liposomes (200 μM total lipids), consisting of DOPC, diphytanoyl-PC, diC16:0-PI4P and Rhod-PE (68/25/5/2 mol/mol) or DOPC, diphytanoyl-PS, diC16:0-PI4P and Rhod-PE (68/25/5/2 mol/mol) were incubated with NBD-$PH_{FAPP}$ (250 nM) in HK buffer at 30 °C. Then DOPC liposomes (200 μM lipids, $L_A$) doped with 5% POPS were added. After 2 min, Osh6p, Osh6pΔ35 or Osh6pΔ69 (200 nM) was injected. The dequenching of the NBD signal mirrors the translocation of NBD-$PH_{FAPP}$ from $L_B$ to $L_A$ liposomes due to the transport of PI4P. The signal is normalized in term of amount of PI4P delivered into $L_A$ liposomes. **b** Initial PI4P transport rates of each Osh6p construct measured with $L_B$ liposomes containing diphytanoyl-PC (orange bar) or –PS (brown bar). The rates, represented as mean ($n = 4$–9), were determined from experiments as shown in (**a**). **c** Time evolution of the binding of NBD-labeled wild-type Osh6p or its Δ35 counterpart to $L_A$ and $L_B$ liposomes. $L_A$ Liposomes, incorporating or not 5% POPS were added to $L_B$ liposomes, comprising 25% of either diphytanoyl-PS or -PC, doped or not with 4% diC16:0-PI4P, in HK buffer at 30 °C. Then, NBD-Osh6p WT or Δ35 was added (200 nM). The fluorescence was followed at 535 nm ($\lambda_{ex} = 460$ nm) over 20 min and normalized. Each trace is representative of several experiments. The dashed line corresponds to the NBD signal measured from the protein in buffer alone and which corresponds to zero after normalization. Error bars correspond to s.e.m. Source data are provided as a Source Data file

onto the membrane via residues of the β17-β18 loop and, transiently, of the β14-β15 loop (Fig. 6c, d; Supplementary Fig. 9a) during a short period (244–365 ns) of the Δ35-1 trajectory (Supplementary Movie 1). The entry of the pocket remains far from the membrane (shortest $h_{entry} = 20.22$ Å, Supplementary Fig. 9c). For Osh6pΔ69, the β11-β12 and β14-β15 loops systematically make contacts with the membrane during a significant time fraction of each simulation (f = 25–50%). During the Δ69-1, -2 and -3 trajectories, the β7-β8, β16-β17 and β17-β18 loops and β18-sheet also frequently insert between lipids (Fig. 6c, d). In the fourth trajectory, Osh6pΔ69 adopts another docking geometry, due to a combined insertion of strands (β7, β8) and loops (β7-β8, β11-β12, β14-β15) in the membrane (Supplementary Fig. 9b). At the end of all simulations, the entry of the pocket is near to the bilayer surface

(shortest $h_{entry} = 4.38$ Å, Supplementary Fig. 9c). Interestingly, during the Δ69-4 trajectory, the entry is in the membrane plane, ($h_{entry}$ between −0.55 and 1 Å), thus in a configuration maybe compatible with an extraction process (Supplementary Fig. 9b, c).

The difference in accessible surface area between Δ35 and Δ69 structures indicates that a part of the lid covers the β7 sheet and β11-β12 loop, which can anchor Osh6pΔ69 in the bilayer (Supplementary Fig. 9d). If Osh6pΔ35 was positioned on membrane surface like Osh6pΔ69, the lid would insert between the lipids (Supplementary Fig. 9e), which is unlikely as it is anionic. This suggests that the lid forbids some docking modes. We examined this by performing a simulation in which Osh6pΔ35 was already anchored to the membrane (final configuration of the Δ35-1 trajectory) but from which the lid was deleted. Remarkably, it comes closer to the membrane plane

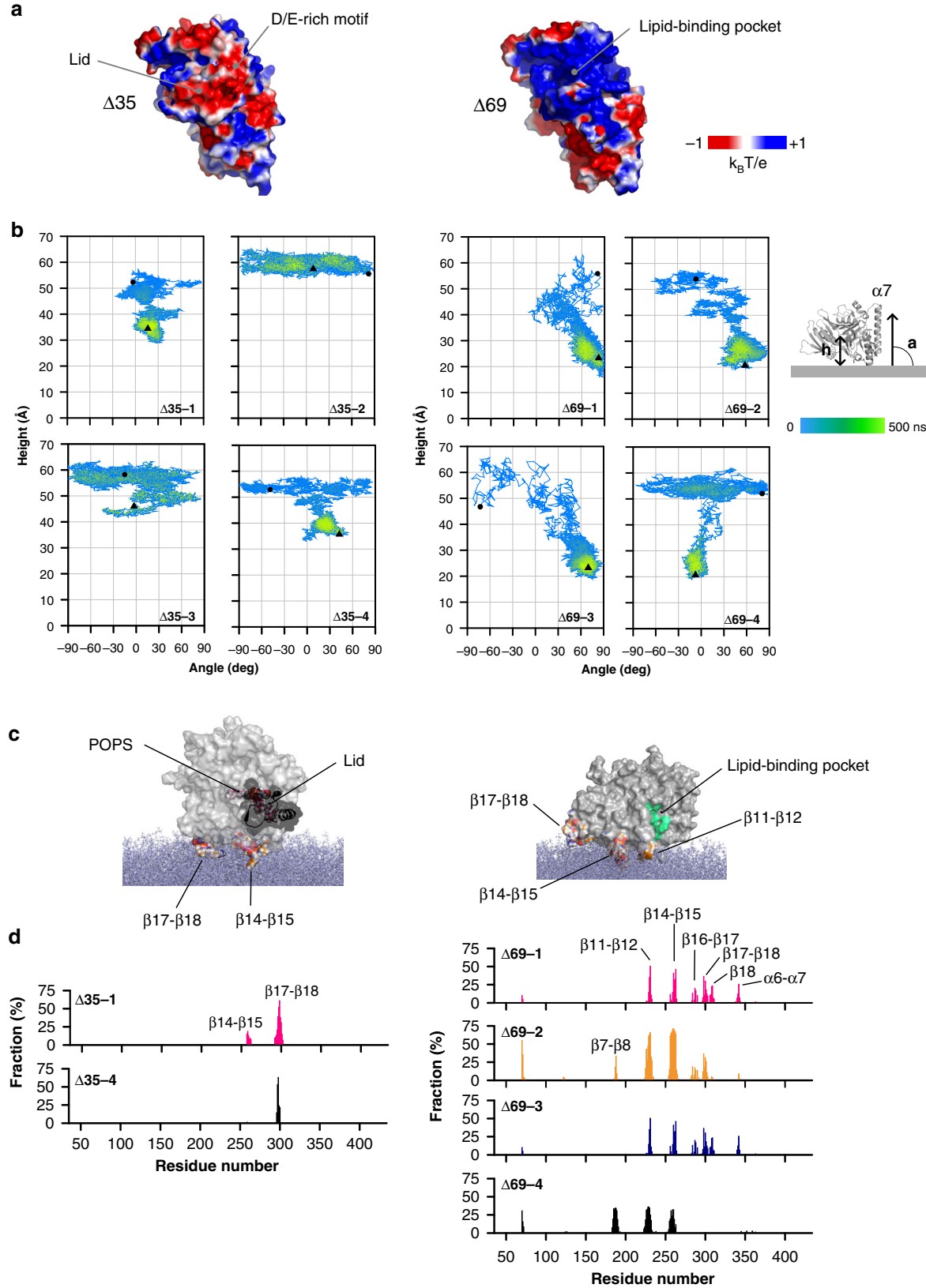

while changing its orientation (Fig. 7a–c), and adopts a configuration akin to that of Osh6pΔ69 at the end of Δ69-1, 2, and 3 trajectories, as indicated by the analysis of the last 100 ns of each simulation (Fig. 7c, d). In summary, our simulations suggest that the lid, when closing, prevents Osh6p from interacting with

membrane because it is anionic and masks some membrane-binding regions.

**Osh6p loses its activity if its lid is less anionic.** We then analyzed to what extent the D/E-rich motif in the lid might play a

**Fig. 6** The lid of Osh6p prevents the protein from binding to PS-rich membranes. **a** Electrostatic features of the molecular surface of Osh6pΔ35 and Osh6pΔ69. **b** Co-evolution of the height and orientation of the protein relative to the membrane surface during each trajectory. The height is calculated between the mass center of Osh6pΔ35 (residue 35–434, trace Δ35-1 to-4) or Osh6pΔ69 (residue 70–434, trace Δ69-1 to 4) and the plane of the DOPC/POPS bilayer during each 500 ns simulation. The orientation is given by the angle between the 346–356 segment of the α7 helix and the plane of the membrane. The time evolution is shown with colors going from blue (beginning of the MD trajectory) to green (end of the trajectory). Dark circles and triangles correspond to the distance and angle values associated to the initial and final configuration ($t = 500$ ns) of each protein, respectively. **c** Configuration of Osh6pΔ35 and Osh6pΔ69, bound to the surface of the lipid bilayer at 300 ns of the Δ35-1 trajectory and at the end of the Δ69-2 trajectory, respectively. The surface of the protein is shown except for residues of the β11–β12, β14–β15, and β17–β18 loops, which are in contact with the membrane: the atoms are represented as sphere with carbon in orange, oxygen in red, nitrogen in blue and hydrogen in gray. The lid with the D/E-rich motif is drawn in a ribbon mode. The molecule of POPS inside the binding cavity of Osh6pΔ35 is represented in a stick mode. Lipids of the membrane are shown in blue grey as stick. The walls of the binding site are colored in green. **d** Percentage of time Osh6p's residues are in close contact with the membrane. The identities of loops and others structural elements that insert in the membrane, are indicated. Source data are provided as a Source Data file

role in Osh6p's activity. To this end, we purified two mutants, Osh6p-4A and Osh6p-5A2G, in which respectively four or seven aspartate and glutamate residues within this motif were substituted by alanine or glycine for neutralizing anionic charges (Fig. 8a). Electrostatic potential calculations show that in the 4A and 5A2G structures, compared to the native one, a part of the lid exposes a positively-charged surface whose area increases with the number of substitutions (Fig. 8b). Both mutants significantly extracted PS or PI4P from liposomes, suggesting that the lid was functional (Fig. 8c). Flotation assays indicated that they interacted more with membrane containing 30% PS and 4% PI4P than Osh6p, suggestive of a tendency to remain adsorbed onto anionic membranes despite the presence of ligands (Fig. 8d). Next, we measured using ER-like ($L_A$) and PM-like ($L_B$) liposomes how the negative charge of membranes affected the activity of these mutants, compared to Osh6p (Fig. 8e). The rate of PI4P transfer mediated by 4A and 5A2G mutants is divided by ~7 and 22, respectively, when $L_B$ liposomes contain 25% diphytanoyl-PS instead of diphytanoyl-PC, more than for Osh6p (~3, Fig. 8f). We obtained similar results (Fig. 8f) with $L_A$ and $L_B$ liposomes that additionally contained PI and PA to better mimic the ER-membrane and the PM[8]. We tested the functionality of the more strongly affected 5A2G mutant in yeast. First we examined how it supported growth of a strain lacking all seven *OSH* genes and only expressing a temperature-sensitive allele of *osh4*. These cells cannot survive at the non-permissive temperature of 37 °C, but grow at 25 °C, where the *osh4-1^{ts}* allele is functional[42]. The growth at 37 °C was rescued by expression of Osh6p (Fig. 8g), likely as it substitutes for Osh4p to remove PI4P from post-Golgi vesicles and thereby restore exocytosis[43]. We also confirmed that the L69D and HH/AA mutants fail to restore growth[43], possibly as they cannot extract and transport PI4P (Fig. 2d and ref. [18]). In contrast, the 5A2G as well as the Δ35 and Δ69 mutants rescue growth. Corroborating in vitro data, this suggests that in yeast Osh6p-5A2G recognizes PI4P, like the deletion mutants. However, whereas Osh6p transfers PS from the ER to the cell surface (Fig. 8h and refs. [15,18]), the expression of Osh6p-5A2G produces an intermediate phenotype: some PS reaches the surface and some remains at the ER during the time-course (Fig. 8h). All these data suggest that attenuating the anionic nature of the lid limits Osh6p in its capacity to escape from negatively-charged membranes and to displace lipids.

## Discussion

Deciphering the mode of action of LTPs, at the core of lipid distribution, is an important issue in cell biology[1,2]. Many models implicitly assume that LTPs bind transiently to organelles to quickly transfer lipids[30,32,33], yet almost nothing is known about these association/dissociation steps. We found that Osh6p, a PS/PI4P exchanger of the ORP/Osh family, has no propensity to

associate with the lipid surface of the ER and the PM and a weak avidity for neutral or negatively-charged liposomes in vitro. In contrast, Osh6p strongly associates with yeast membranes, including the ER, and with anionic liposomes, if the lid that controls the entry of its lipid-binding pocket is absent or unable to close. It also binds to anionic membranes if they are devoid of extractable ligands, PS or PI4P. If these lipids are available, Osh6p dissociates because its lid closes upon lipid extraction. These data supports the idea that the occupancy of a LTP by its ligand can modulate its conformational state and therefore its membrane-binding capacity[17,33]. Interestingly, this resembles the mechanism that controls the recruitment of small G-proteins onto membranes depending on their ligand status (GDP or GTP-bound)[44,45]. Using a model system mimicking the ER/PM interface, we established that a high density of anionic lipids in PM-like membrane weakly impacts the transport activity of Osh6p and its Δ35 counterpart, because both have a functional lid. During this process, they remain largely soluble. The lid is thus a structural element that allows Osh6p to stay very briefly on the membrane, and thereby to ensure lipid exchange, regardless of the density of anionic lipids.

To understand why, we performed in silico analyses on the structure of Osh6p, i.e., Osh6pΔ35[15], and of the lidless Osh6pΔ69, taking into account several points. First, Osh6pΔ35 behaves like Osh6p: the 1–35 region upstream of the lid plays no role in the coupling between lipid extraction and membrane dissociation. Second, Osh6pΔ69 is equivalent to Osh6p(L69D), thus to a constitutively open form of Osh6p. Third, modeling a structure of Osh6p with an open lid was problematic. These analyses indicated that the lid is anionic, due to a D/E-rich motif, and that it changes the electrostatic features of Osh6p's surface when closed. In MD simulations, Osh6pΔ35 binds less to an anionic membrane than Osh6pΔ69, due to weaker non-covalent interactions. Osh6pΔ35 inserts in the bilayer via its β17-β18 loop and, sometimes, its β14-β15 loop. Interestingly, a corresponding loop seems critical for anchoring Osh4p to membrane[46]. In comparison, the insertion of Osh6pΔ69 relies on these loops but also on extra regions, including a positively-charged surface surrounding the entry of the pocket that is exposed when the lid is absent. This entry can localize at the water/membrane interface, suggesting that Osh6p, once open, adopts a configuration compatible with an extraction/delivery process. Finally we showed that charge-neutralizing mutations of the D/E-rich motif increase the ability of Osh6p to bind to anionic membranes and can lower its transfer activity in vitro and at the ER/PM interface.

We propose that the lid supports an electrostatic switching mechanism that helps Osh6p to transiently associate with the ER and the PM. Thus Osh6p can freely circulate between these compartments to ensure continuous lipid exchange. This would explain why a large fraction of this LTP is cytosolic at steady-state and why the L69D and Δ69 mutants, with no functional lid and a

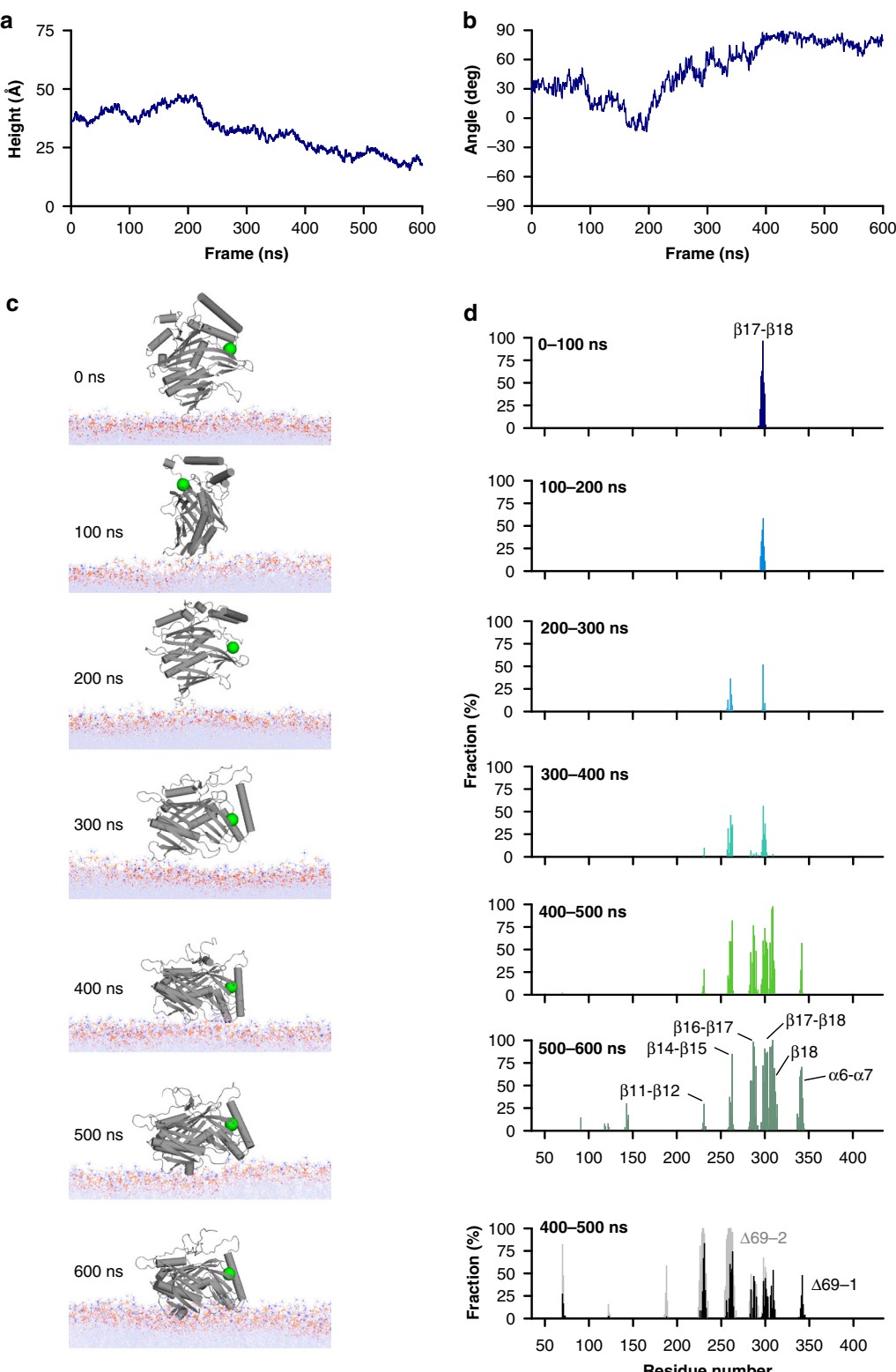

**Fig. 7** Removal of the lid elicits the reorientation of Osh6p on a bilayer surface. **a** Evolution of the height between the center of mass of the protein relative to the plane of the bilayer during 600 ns. **b** Evolution of the angle between the α7-helix and the plane of the membrane during the trajectory. **c** Snapshot at time zero and every 100 ns of the trajectory showing the reorientation of the protein relative to the lipid bilayer. **d** Percentage of time Osh6p's residues (f values) are in contact with membrane during six successive periods (100 ns each) of the trajectory. The loops corresponding to residues in contact with membrane are indicated. As a comparison, the f values are indicated for the last 100 ns of the Δ69-1 and Δ69-2 trajectories, during which Osh6pΔ69 is membrane-bound. The f values of the last 100 ns of the Δ69-3 trajectory, similar to those of the Δ69-2 trajectory, were omitted for clarity. Source data are provided as a Source Data file

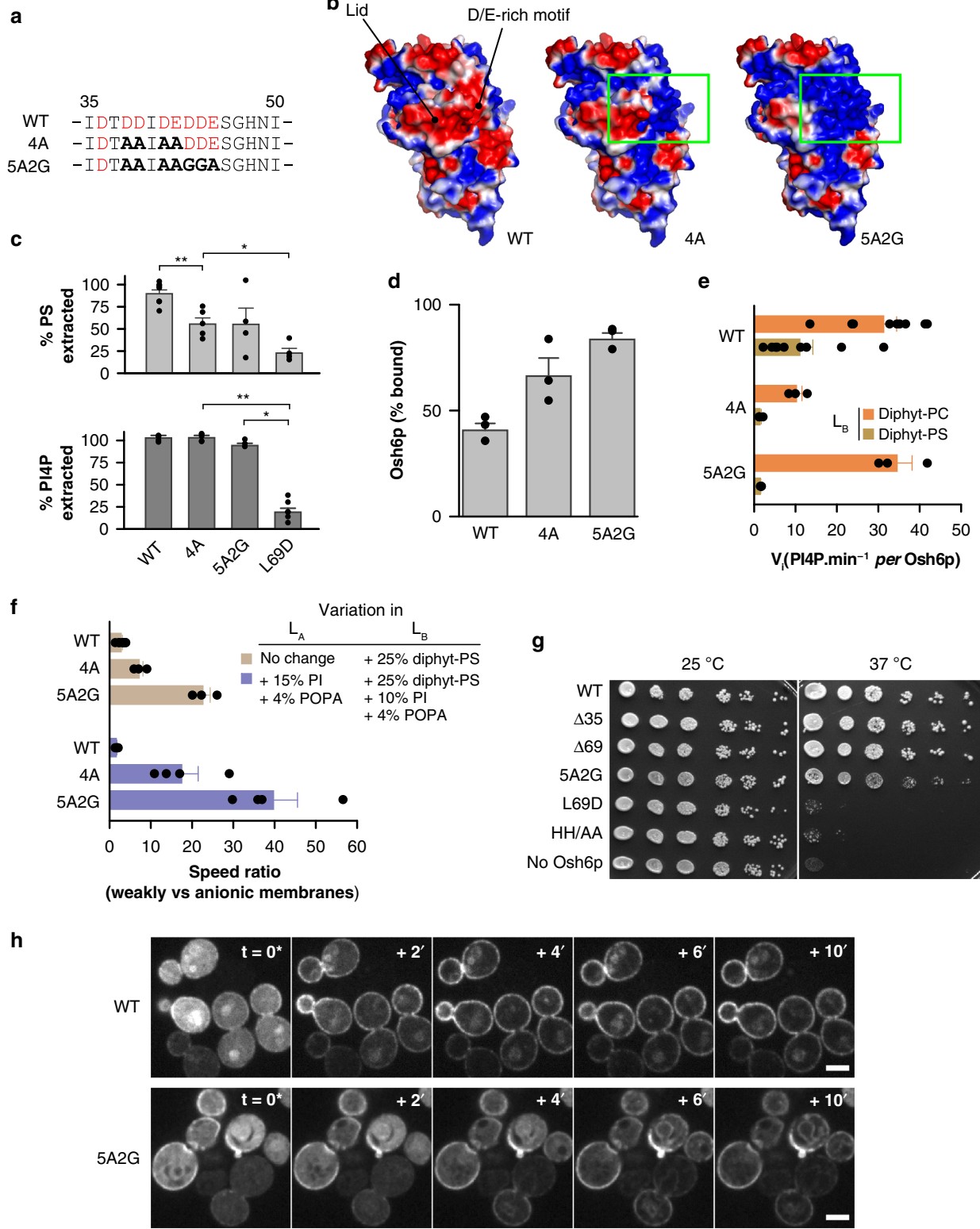

higher propensity to interact with anionic surfaces, display an increased localization to intracellular membranes. The precision in PS/PI4P exchange mediated by Osh6p can be therefore assured by continuous PS synthesis at the ER and the maintenance of a PI4P gradient at the ER/PM interface[18]. Presumably, additional factors contribute to the targeting of Osh6-mediated lipid transport, including protein–protein interactions. However, the accuracy of transport likely does not rely on the ability of Osh6p to

differentially bind organelle surfaces according to their bulk lipid composition. When its ligands are present, the avidity of Osh6p for membranes and its activity remain largely insensitive to the density (up to 45%) and nature of anionic lipids. Nevertheless, the 1–35 region, upstream of the lid, might have targeting properties. We noted subtle differences between Osh6p and Osh6pΔ35 in vitro: Osh6p keeps some avidity for anionic membranes containing PI4P and its PI4P transport activity is slightly more

**Fig. 8** Mutations in the D/E-rich motif impact Osh6p activity. **a** Sequence of the D/E-rich motif with the anionic residues colored in red. Substitutions by alanines or glycines in the 4A and 5A2G mutant are in bold. **b** Electrostatic features of the surface of Osh6p and mutants. **c** Percentage of accessible PS or PI4P extracted by Osh6p, Osh6p-4A or Osh6p-5A2G (3 μM) from liposomes (80 μM lipids) with 2% POPS or diC16:0-PI4P ($n = 3$–8, $*P < 0.05$, $**P < 0.01$, Mann–Whitney test). **d** Flotation assay. Osh6p, Osh6p-4A or Osh6p-5A2G was mixed with DOPC/POPS/diC16:0-PI4P (66/30/4 mol/mol) liposomes ($n = 3$). **e** Initial PI4P transport rates of Osh6p and of 4A and 5A2G mutants (200 nM) determined with $L_B$ liposomes containing 5% diC16:0-PI4P and either 25% diphytanoyl-PC (orange bar) or –PS (brown bar) and $L_A$ liposomes containing 5% POPS ($n = 3$–9). **f** Ratio between the initial PI4P transport rate, measured for Osh6p and mutants when $L_A$ and $L_B$ liposomes are weakly anionic (with 5% POPS and 5% PI4P, respectively) and the transport rate measured with $L_B$ liposomes that are more anionic (with 25% of diphytanoyl-PS instead of diphytanoyl-PC (as in (**e**), brown bar, $n = 3$–9) or the rate measured with $L_A$ and $L_B$ liposomes that are both more anionic, doped respectively with 15% PI and 4% POPA (at the expense of DOPC), and 10% PI, 4% POPA and 25% diphytanoyl-PS instead of diphytanoyl-PC (purple bars, $n = 4$). **g** Growth of $osh1\Delta$-$7\Delta$ yeast, containing a plasmid-borne copy of a temperature-sensitive allele of OSH4 and the indicated version of OSH6, at permissive (25 °C) or non-permissive (37 °C) temperature for $osh4$-$1^{ts}$. **h** Redistribution of C2$_{Lact}$-GFP in PS-depleted yeast cells ($cho1\Delta osh6\Delta osh7\Delta$) expressing Osh6p-mCherry or the 5A2G mutant after adding 18:1 lyso-PS. Images were taken every 2 min, with the first panel ($t = 0^*$) showing the last time-point before the onset of C2$_{Lact}$-GFP signal transition. The absolute timing of this time-point varied between experiments (15–25 min after lyso-PS injection) due to instability of the lyso-PS suspension[18]. Scale bar: 5 μm. Error bars correspond to s.e.m. Source data are provided as a Source Data file

---

sensitive to the presence of anionic lipids, compared to Osh6pΔ35. The 1–35 region, with a net charge of +4, might allow Osh6p to target phosphoinositides; unfolded positively-charged motifs, used by proteins like MARCK to interact with anionic bilayers, can recognize PI(4,5)P$_2$[47].

It is unknown whether other ORP/Osh proteins utilize a switching mechanism like Osh6p. The membrane-binding capacity of Osh6p depends on whether or not the lid occludes a basic surface at the entry of the binding pocket, which is well-conserved among ORP/Osh proteins. One might speculate that in Osh4p, whose lid has no D/E-rich motif[17,48], this occlusion alone modulates its association with the ER and Golgi/post-Golgi membranes, which are less anionic than the PM and between which this protein operates[49–51]. ORP5 and ORP8 are PS/PI4P exchangers at ER-PM contact sites[20], but their ORD has no D/E-rich motif (Supplementary Fig. 5), in contrast to ORP10, which exchanges PS and PI4P at ER-Golgi contacts[52]. These proteins are more complex as they reside in the ER and bind to the Golgi or PM via a PH domain[20]. Possibly, in that configuration, the ORD targets sequentially the membranes through weak interactions and a D/E-rich motif is not critical. In contrast, Osh6p is unique among the PS/PI4P exchangers because its ORD must both target membrane and exchange lipids. Consequently, the anionic nature of the lid would be necessary to break the interaction of the ORD with the membrane, once lipids are exchanged. This might be even critical in a cellular context. Indeed, if Osh6p remains empty, it aggregates membranes. This is partially reminiscent of a previous study showing that Osh6p strongly bound to and aggregated liposomes although they contained PS[40]. However, in this study, the liposomes were not representative of cellular membranes because they were very anionic (50–100% DOPS) with a loose lipid packing; likely, these combined features artificially promoted protein-membrane interactions[22]. Interestingly, like Osh6p, the membrane-binding capacity of PITPα, a member of a distinct LTP family, depends on whether or not it hosts its ligands[36]. It is possible that several ORP/Osh proteins and many other types of LTPs have a switching mechanism whose nature remains to be defined.

## Methods

**Protein expression and purification**. Osh6p was cloned in pGEX-4T-3 vector to code for a GST-fused construct[18]. The different site-specific mutations or deletions in Osh6p were obtained by the Quikchange kit (Agilent). The oligonucleotides are listed in the Supplementary Table 1. For the truncated [70–448]Osh6p mutant (Osh6pΔ69), a short GGGG linker was introduced between the protein and GST to facilitate the cleavage by thrombin[18]. To build the Osh6p(noC/S190C), the sequence of the protein was mutated to replace three endogenous cysteine residues (C62, C162, C389) by serine and to substitute a serine located under the lid by a cysteine (S190C mutation). To prepare NBD-labeled Osh6p WT and NBD-labeled Osh6pΔ35, the endogenous cysteines were replaced by serines and a cysteine was

introduced into the β14–β15 loop by mutating a threonine (T262C substitution). To build the mutant Osh6p-4A, four anionic residues in Osh6p WT (D38, D39, D41, and E42) were replaced each by an alanine residue. In Osh6p-5A2G, three additional mutations were performed (D43G, D44G, E45A). Glycine residues were used instead of alanine residues to limit the propensity of the mutated segment to fold into a continuous α-helix. The design of the C2$_{Lact}$ and PH$_{FAPP}$ constructs and the expression system have been previously described[18,41]. All the mutations or deletions were checked by DNA sequencing.

GST-Osh6p and mutants were expressed in *E. coli* (BL21-Gold(DE3) competent cells, Stratagene) at 30 °C overnight whereas the GST-PH$_{FAPP}$ and GST-C2$_{Lact}$ were expressed at 37 °C for 3 h upon induction with 1 mM IPTG (when the optical density (OD) of the bacterial suspension, measured at 600 nm, reached a value of 0.6). All purification steps were conducted in a buffer with 50 mM Tris, pH 7.4, 120 mM NaCl, 2 mM DTT, supplemented during the first purification steps with 1 mM PMSF, 1 μM bestatine, 10 μM pepstatine, 10 μM phosphoramidon and protease inhibitor tablets (Roche). Cells were lysed by a French press and the lysate was centrifuged at 200,000 × g for 1 h. The supernatant was applied to Glutathione Sepharose 4B beads. After three washing steps, the beads were incubated with thrombin at 4 °C overnight to cleave off the GST tag and release the protein. Afterward, they were purified by gel filtration on a Sephacryl S200 HR XK16-70 column. All constructs contain an N-terminal GS sequence from the thrombin cleavage site.

For NBD labeling of Osh6p, PH$_{FAPP}$ or C2$_{Lact}$, the crude eluate obtained after thrombine treatment was concentrated and mixed (after DTT removal by gel filtration on illustra NAP-10 columns (GE Healthcare)) with a 10-fold excess of N, N′-dimethyl-N-(iodoacetyl)-N′-(7-nitrobenz-2-oxa-1,3-diazol-4-yl) ethylenediamine (IANBD-amide, Molecular Probes). After 90 min on ice, the reaction was stopped by adding a 10-fold excess of L-cysteine over the probe. The free probe was removed by gel filtration on a Sephacryl S200 HR XK16-70 column. The labeled protein was analyzed by SDS-PAGE and UV-visible spectroscopy. The gel was directly visualized in a fluorescence imaging system (FUSION FX fluorescence imaging system) to detect NBD-labeled protein excited in near-UV and then stained with Sypro Orange to determine the purity of NBD-labeled protein. The labeling yield (≈100%) was estimated from the ratio of the OD of tyrosine and tryptophan at 280 nm ($\varepsilon = 29,450$ M$^{-1}$ cm$^{-1}$ for PH$_{FAPP}$, $\varepsilon = 45,045$ M$^{-1}$ cm$^{-1}$ for C2$_{Lact}$) and NBD at 495 nm ($\varepsilon = 25,000$ M$^{-1}$ cm$^{-1}$). For all purified proteins, the concentration was determined by a BCA assay, by SDS-PAGE analysis using a BSA standard curve and UV spectrometry.

**Liposomes preparation**. DOPC (1,2-dioleoyl-*sn*-glycero-3-phosphocholine), POPC (1-palmitoyl-2-oleoyl-*sn*-glycero-3-phosphocholine), diphytanoyl-PC (1,2-diphytanoyl-*sn*-glycero-3-phosphocholine), diC12:0-PS (1,2-dilauroyl-*sn*-glycero-3-phospho-L-serine), POPS (1-palmitoyl-2-oleoyl-*sn*-glycero-3-phosphoserine), diphytanoyl-PS (1,2-diphytanoyl-*sn*-glycero-3-phospho-L-serine), POPA (1-palmitoyl-2-oleoyl-*sn*-glycero-3-phosphate), liver PI (L-α-phosphatidylinositol), NBD-PE (1,2-dioleoyl-*sn*-glycero-3-phosphoethanolamine-N-(7-nitro-2-1,3-benzoxadiazol-4-yl)), Rhod-PE (1,2-dipalmitoyl-*sn*-glycero-3-phosphoethanolamine-N-(lissamine rhodamine B sulfonyl)) were purchased from Avanti Polar Lipids. diC16:0-PI4P (1,2-dipalmitoyl-*sn*-glycero-3-phosphoinositol-4-phosphate) was from Echelon Lipids Inc.

Lipids in stock solutions in CHCl$_3$ were mixed at the desired molar ratio. The solvent was removed in a rotary evaporator. For lipid films doped with diC16:0-PI4P the mix was pre-warmed at 33 °C for 5 min prior to drying under vacuum. The films were then hydrated in 50 mM HEPES, pH 7.2, 120 mM K-acetate (HK buffer) to obtain a suspension of multilamellar liposomes. After five thawing-freezing cycles, the suspensions were extruded through polycarbonate filters of 0.2 μm pore size using a mini-extruder (Avanti Polar Lipids). Mean hydrodynamic radius and polydispersity of liposomes were determined by dynamic light scattering at 25 °C using a Dynapro apparatus (Protein Solutions). Each sample of extruded

                                                                                                                                                       

liposomes was diluted at 50 µM in HK buffer in a 20 µL quartz cell. A set of 12 autocorrelation curves was acquired and analyzed using a regularization algorithm. Representative mean radius and polydispersity of liposomes, used in this study, according to their lipid composition, are listed in the Supplementary Table 2. Liposomes were stored at 4 °C and in the dark when containing fluorescent lipids and used within 2 days.

**Flotation experiment**. The protocol has been described in detail[53]. Briefly, wild-type Osh6p or each mutant (0.75 µM) was mixed with NBD-PE containing liposomes (750 µM total lipids) in 150 µL of HK buffer at 25 °C for 10 min under agitation. The suspension was adjusted to 28% (w/w) sucrose by mixing 100 µL of 60% (w/w) sucrose solution in HK buffer and overlaid with 200 µL of HK buffer containing 24% (w/w) sucrose and 50 µL of sucrose-free HK. The sample was centrifuged at 240,000 × g in a swing rotor (TLS 55) for 1 h. The bottom (250 µL), middle (150 µL) and top (100 µL) fractions were collected. The bottom and top fractions were analyzed by SDS-PAGE using Sypro-Orange staining and a FUSION FX fluorescence imaging system.

**PS and PI4P extraction assay**. Measurements were done in a 96-well black plate using a TECAN M1000 Pro. For PS extraction assays, DOPC liposomes (80 µM total lipids) doped with 2% POPS (0.8 µM accessible concentration) were mixed with NBD-C2$_{Lact}$ (250 nM) at 25 °C in the presence or absence of 3 µM Osh6p or mutants in individual well (100 µL final volume). NBD spectra were recorded from 505 to 650 nm (bandwidth 5 nm) upon excitation at 490 nm (bandwidth 5 nm). The contribution of buffer or liposomes alone was subtracted from the NBD signal. A control spectrum was measured with the NBD-C2$_{Lact}$ mixed with DOPC liposomes devoid of PS. The intensity at 536 nm measured with PS-containing liposomes in the absence or presence of the protein corresponds to $F_{max}$ and $F$, respectively, whereas the intensity measured at the same wavelength with pure DOPC liposomes corresponds to $F_0$. The percentage of PS extraction is given by using the formula $100 \times (1 - ((F - F_0)/(F_{max} - F_0)))$.

For PI4P extraction assays, liposomes doped with 2% diC16:0-PI4P were prepared and measurements were done with the NBD-PH$_{FAPP}$ probe. Liposome and protein concentrations were identical to those used in the PS extraction assays. Control experiments and determination of extraction percentage were performed similarly.

**CPM accessibility assay**. A stock solution of CPM (7-Diethylamino-3-(4-maleimidophenyl)-4-methylcoumarin, Sigma-Aldrich) at 4 mg.mL$^{-1}$ was prepared as in ref. [54] by mixing 1 mg of CPM powder in 250 µL of DMSO. Thereafter, this solution was diluted in a final volume of 10 mL of HK buffer and incubated for 5 min at room temperature. The solution was protected from light and used immediately. Osh6p(noC/S190C) or its L69D counterpart (400 nM) was mixed at 30 °C with liposomes of various composition (400 µM total lipids) in 200 µL of HK buffer. A small volume of the CPM stock solution was added to obtain a final concentration of 4 µM. Emission fluorescence spectra were measured from 400 to 550 nm (bandwidth 5 nm) upon excitation at 387 nm (bandwidth 5 nm) in a 96-well plate using a TECAN M1000 Pro. The intensity value at 450–470 nm corresponds to the spectral peak. Control spectra were recorded in the absence of protein.

**NBD-based membrane binding assay**. NBD-labeled Osh6p WT or Δ35 (200 nM) was incubated with liposomes of various composition (400 µM total lipids) in HK buffer for 10 min at 25 °C. Emission fluorescence spectra were recorded from 500 to 650 nm (bandwidth 5 nm) upon excitation at 460 nm (bandwidth 5 nm) in a 96-well plate using a TECAN M1000 Pro. Control spectra were measured in the absence of protein to subtract light scattering from liposomes.

**Real-time membrane dissociation assay**. NBD-labeled Osh6p WT or Δ35 (100 nM) was mixed with DOPC/diphytanoyl-PS liposomes (70/30 mol/mol, 400 µM lipid concentration) at 30 °C in a cuvette under constant stirring in HK buffer. Then, 1 µL of a stock solution of diC12:0-PS in methanol (12 mM) was injected to obtain a final concentration of 20 µM. Emission fluorescence over time was measured at 535 nm (bandwidth 5 nm) upon excitation at 460 nm (bandwidth 5 nm). A control measurement was done by injecting methanol alone. As a comparison, the experiment was repeated with NBD-C2$_{Lact}$ (250 nM) incubated with liposomes. These experiments were performed with a Shimadzu RF 5301-PC spectrofluorometer.

**PI4P transport assay**. This assay was carried out in a Shimadzu RF 5301-PC spectrofluorometer. The sample (volume 600 µL) was placed in a cylindrical quartz cell, continuously stirred with a small magnetic bar and thermostated at 30 °C. At the indicated time, samples were injected from stock solutions with Hamilton syringes through a guide in the cover of the fluorometer. A time zero, a suspension (570 µL) of L$_B$ liposomes (200 µM total lipids, final concentration) containing 2% Rhod-PE and 5% PI4P was mixed with 250 nM NBD-PH$_{FAPP}$ in HK buffer. The volume concentration of accessible PI4P (in the outer leaflet) was 5 µM. After 2 min, 30 µL of a suspension of L$_A$ liposomes (200 µM lipids), incorporating 5% POPS, were injected. After two additional minutes, Osh6p (200 nM) was injected. PI4P transport was followed by measuring the NBD signal at 530 nm (bandwidth 10 nm) upon excitation at 460 nm (bandwidth 1.5 nm). This signal reflects the repartition of NBD-PH$_{FAPP}$ between L$_B$ and L$_A$ liposomes. The amount of PI4P transported by Osh6p can be determined by a normalization of the NBD signal through the following procedure. First we measured the NBD signal ($F_{max}$) corresponding to a situation where PI4P is fully transferred to L$_A$ liposomes. NBD-PH$_{FAPP}$ (250 nM) was mixed with L$_A$ and L$_B$ liposomes (200 µM total lipid each) with a lipid composition similar to that of the liposomes used in the transport assay, except that L$_A$ liposomes initially contained 5% PI4P whereas L$_B$ liposomes were devoid of PI4P. The fraction ($F_{Norm}$) of PI4P in L$_A$ liposomes, PI4P$_A$/PI4P$_{T}$, is directly equal to the fraction of PH$_{FAPP}$ bound to L$_A$ liposomes and is calculated by considering $F_{Norm} = (F - F_0/F_{max} - F_0)$ with $F_0$ corresponding to the NBD signal prior to the addition of Osh protein. The amount of PI4P (in µM) transferred from L$_B$ to L$_A$ liposomes corresponds to $5 \times F_{Norm}$. The initial transport rates were determined from normalized curves by fitting with a linear function the first eight data points, corresponding to a period of 4 s, measured just after Osh6p injection.

To examine whether Osh6p or its Δ35 counterpart was bound or not to L$_A$ and L$_B$ liposomes during PI4P transport assays, we performed the same experiments as described above but in the absence of NBD-PH$_{FAPP}$ and by injecting the NBD-labeled form of Osh6p and Osh6pΔ35 (200 nM) instead of the unlabeled forms. Additional experiments were done with L$_A$ and L$_B$ liposomes devoid of PS and PI4P, respectively. The percentage of Osh6p bound to liposomes corresponds to $100 \times (F - F_{min}/F_{max} - F_{min})$ with $F_{min}$ and $F_{max}$ corresponding respectively to the signal of the NBD-labeled Osh6p proteins in HK buffer only or in the presence of liposome (400 µM total liposomes) containing 30 mol% diphytanoyl-PS.

**Circular dichroism**. The experiments were performed on a Jasco J-815 spectrometer at room temperature with a quartz cell of 0.05 cm path length. Osh6p was dialysed against a buffer with 20 mM Tris, pH 7.4, 120 mM NaF to remove glycerol. Each spectrum is the average of several scans recorded from 185 to 260 nm with a bandwidth of 1 nm, a step size of 0.5 nm and a scan speed of 50 nm min$^{-1}$. A control spectrum of buffer was subtracted from each protein spectrum. The spectra were analyzed in the 185–240 nm range using the SELCON3 method provided on-line by the DICROWEB server[55].

**Yeast strains and plasmids and manipulations**. Yeast manipulations were done using standard methods and growth conditions. Strain Osh6-GFP, with the genotype MATa his3Δ leu2Δ0 met15Δ0 ura3Δ0 OSH6-GFP::HIS3MX6, was from the GFP collection[56] supplied by S Léon. All other strains were in the BY4741 background, except for the osh1Δ-7Δ osh4-1$^{ts}$ strain: strains osh6Δ osh7Δ and cho1Δosh6Δosh7Δ were a gift from AC Gavin, with genotypes MATa can1Δ::STE2pr-LEU2 lyp1Δ his3Δ leu2Δ0 met15Δ0 ura3Δ0 osh6Δ::HYGMX osh7Δ::NATMX, and MATa can1Δ::STE2pr-LEU2 lyp1Δ his3Δ leu2Δ0 met15Δ0 ura3Δ0 cho1Δ::KANMX osh6Δ::HYGMX osh7Δ::NATMX ([15]). The osh1Δ-7Δ osh4-1$^{ts}$ strain, CBY926, which is in the SEY6210 background, with the genotype MATα ura3-52 leu2-3,112 his3Δ200 trp1-Δ901 lys2-801 suc2Δ9 osh1Δ::KANMX osh2Δ::KANMX osh3Δ::LYS2 osh4Δ::HIS3 osh5Δ::LEU2 osh6Δ::LEU2 osh7Δ::HIS3 posh4-1$^{ts}$ TRP1 CEN, was a gift from C. Beh, supplied by CL Jackson[42]. Yeast plasmids used in this study are listed in Supplementary Table 3. The plasmid pRS315-Osh6-mCherry was a gift from AC Gavin, the Sec63-sGFP plasmid was obtained from Addgene (#8854) and the plasmid pRS416-C2$_{Lact}$-GFP was from Hematologic technologies. Plasmid pRS315-Osh6-GFP was generated by replacing mCherry with PCR-amplified GFP coding sequence. Plasmids expressing N-terminal truncations of Osh6p were generated by PCR amplification of the corresponding OSH6 fragments and ligation into the pRS315-Osh6-mCherry plasmid digested SpeI-BamHI. Osh6p-expressing plasmids with the URA3 marker (for growth complementation assay in the osh1Δ-7Δ osh4-1$^{ts}$ strain) were generated by swapping LEU2 with the URA3 marker in the pOsh6-mCherry plasmids. The plasmid pOsh6(5A2G)-U was generated by PCR-amplification of the Osh6p(5A2G) coding sequence from the bacterial expression plasmid, and ligation into pOsh6-mCherry-U digested by SpeI and BamHI. The plasmid pC2$_{Lact}$-GFP-L was generated by swapping URA3 with the LEU2 marker in pC2$_{Lact}$-GFP. All plasmid sequences were verified by sequencing.

**Yeast growth assay**. The osh1Δ-7Δ osh4-1$^{ts}$ strain was transformed with the indicated Osh6-mCherry-URA plasmid and the resulting transformants were grown overnight in SC without uracil and tryptophan liquid cultures to stationary phase. They were spotted in 10-fold serial dilutions onto synthetic media plates without uracil and tryptophan and the plates were incubated for 3–4 days at 25 °C and at 37 °C.

**Fluorescence microscopy**. Yeast strains were grown overnight at 30 °C in appropriate SD medium to maintain plasmid selection. For strains carrying the cho1Δ mutation, SD medium was supplemented with 1 mM ethanolamine. Cells were harvested in mid-logarithmic phase (OD$_{600}$ = 0.6–0.8) and prepared for viewing on glass slides when assessing protein localization at steady state, or in a microfluidics chamber in the case of time-course experiments using lyso-PS (see below).

The cellular localization of PS was monitored in *cho1Δ osh6Δ osh7Δ* strains after the addition of exogenous 18:1 lyso-PS (1-oleoyl-2-hydroxy-*sn*-glycero-3-phospho-L-serine, Avanti Polar Lipids) as was previously described[18]. For lyso-PS time course experiments, cells were injected into a YO4C microfluidics chamber using the Microfluidic Perfusion Platform (ONIX), driven by the interface software ONIX-FG-SW (CellASIC Corp.). Cells were trapped and maintained in a uniform focal plane. Normal growth conditions were maintained by flowing cells with SD medium or SD medium containing lyso-PS at 3 psi. The microfluidics device was coupled to a DMI6000 (Leica) microscope, equipped with an oil immersion plan apochromat 100x objective NA 1.4, a sCMOS PRIME 95 (Photometrics), and a spinning-disk confocal system CSU22 (Yokogawa). GFP-tagged proteins and mCherry-tagged proteins were visualized with a GFP Filter 535AF45, and RFP Filter 590DF35, respectively. Images were acquired with MetaMorph 7 software (Molecular Devices). For the lyso-PS time-course assays, we imaged cells every 2 min over a total time of 30–40 min, starting with the time when lyso-PS-containing medium was injected into the system. We always imaged cells in 5 z-sections separated by 0.7 μm, afterwards manually selecting for the best focal plane, in order to correct for any focal drift during the experiment. Usually we imaged cells in four different fields of 512 × 512 pixels. Images were processed with ImageJ (NIH) and with Canvas Draw (canvas X) for levels.

**MD simulation.** All MD simulations were performed using the CHARMM36 force field[57,58] and the GROMACS 5.0 MD simulation engine[59]. Bonds involving hydrogen atoms were constrained using the LINCS algorithm[60] and the integration time step was set to 2 fs. The Nose-Hoover thermostat[61] was used to keep a temperature at 303 °K with a coupling time constant of 1 ps in order to preserve the fluidity of the membrane. For simulations with constant pressure, the Parrinello-Rahman barostat[62] was used to maintain a pressure of 1 bar with a compressibility of $4.5 \times 10^{-5}$ and a coupling time constant of 5 ps. The cell box size was allowed to vary semi-isotropically ($X = Y$ but not $Z$) in all membrane simulations whereas an isotropic pressure coupling was used in the simulation with the protein only. Van der Waals (VDW) interactions were switched to zero over 10 to 12 Å and electrostatic interactions were evaluated using the Particle Mesh Ewald[63]. A bilayer of 1000 lipids with a binary mix of DOPC and POPS (70 and 30%, respectively) was assembled using the CHARMM-GUI membrane builder[64,65] and the CHARMM36 lipid force field[58]. Both leaflets of the membrane were solvated with a 10 nm layer of TIP3P water containing sodium counterions (300 in total considering the two layers)[66]. The final cell box contains 745,971 atoms and its dimensions were ~18.1 × 18.1 × 22.4 nm. The box was relaxed and equilibrated for 50 ns using the equilibration procedure[65] designed for the CHARMM-GUI and using the GROMACS simulation engine. X-ray structure of Osh6p (PDB entry: 4B2Z[15]; since the 1–35 residues are not solved in that structure, this latter was named Osh6pΔ35 in the text), encapsulating a POPS molecule, was placed in one water layer, with a given orientation, at least 50 Å away from the membrane surface, using Pymol (The PyMOL Molecular Graphics System, Version 1.8 Schrödinger, LLC.) TIP3P molecules and salt ions that overlap spatially with the protein structure, or were close, (within 5 Å in distance) were removed. The system was thereafter minimized with GROMACS and equilibrated with 120 mM NaCl using the CHARMM-GUI procedure. Four distinct protein-membrane systems (approximatively 737,820 atoms), in which Osh6pΔ35 has a distinct orientation with respect to the membrane surface, were built, in order to launch four independent 500 ns trajectories (Δ35-1 to 4). Four systems were also built with Osh6pΔ69 structure (~737,350 atoms), which was obtained by deleting the 35-69 region and the POPS molecule from the structure of Osh6pΔ35 and after an equilibration by MD for 40 ns in water. Corresponding trajectories are called Δ69-1 to 4.

**MD analysis.** Classical data analysis Root Mean Square Deviation (RMSD), Root Mean Square Fluctuation (RMSF), and secondary structure analysis were done using GROMACS 5[59] utilities. All molecular pictures and movies were made with Pymol (http://pymol.org/)[67] and Visual Molecular Dynamics (VMD)[68] softwares. Several non-standard analyses were performed to analyze the interaction between Osh6p and the lipid membrane using scripts based on Python language and dedicated functions in the MDAnalysis package[69]. Distance and angle measurements were carried out on the different MD trajectories every 100 ps, resulting in 5000 data values for each parameter and each MD run. For each frame, we calculated and stored the height of Osh6p relative to the membrane, considering the center of mass (COM) of the protein (Osh6pΔ35 or Osh6pΔ69) and the z plane of the membrane (z average value of the nitrogen atoms (named N) of upper lipids). For calculating the height of the entry of the lipid-pocket relative to the z plane of the membrane, we determined a mass center considering the N129, F185 and S190 residues of Osh6p. To calculate the orientation of the protein relative to the membrane, we computed the angle between the 346–356 segment of the α7 helix and the plane of the lipid patch. We also determined the percentage of time Osh6p's residues were in contact with the membrane, by calculating the z distance between the center of mass of each residue and the z plane of membrane (Natoms). For each frame, if the distance was less than a threshold value of 5 Å, we counted it as a contact and added it in the residues matrix. The percentage of contact was calculated for each residue by dividing the number of positive count by the total number of frames. Electrostatic and VDW energies interaction were computed

between accessible and solvent-exposed residues of Osh6p and lipids using the rerun GROMACS tool.

**Structural analysis.** Accessible surface area (ASA) was evaluated on Osh6p's structures with the program NACCESS[70] using a probe radius of 1.5 Å. The electrostatic surfaces of Osh6pΔ35 and all mutants were calculated with the APBS (Adaptive Poisson-Boltzmann Solver) plugin of PyMOL software[67].

**Sequence alignment.** A first complete set of protein sequences belonging to the Oxysterol-Binding Protein family was retrieved either from the Pfam (PF01237) or Interpro (IPR000648) database, considering eukaryotic species only[71,72]. Fragments coming from incomplete genome assembly were ignored and protein sequences that do not contain the EQTSHHPP signature motif (Prosite pattern PS01013) of the ORP/Osh family were rejected[73]. In the case of hypothetical splicing variants, only the full-length coding sequence was considered for the analysis. Furthermore, redundancy amongst the sequences was eliminated by checking available annotations and genomic data. This procedure allowed generating a set of 462 sequences belonging to 83 eukaryotic species. A first multiple sequence alignment with ClustalO[73] was done and sequences that contained the LPTFILE pattern, i.e., the PS-recognizing motif initially identified in the lid of Osh6p[15], were selected. Then, a second alignment was performed (two iterations, without mbed-like clustering) using this subset, which contains 111 sequences belonging to 73 species. Alignments were subsequently analyzed with Jalview[74]. Amino-acid evolution rate was calculated using Consurf upon 4PH7_A PDB structure by maximum likelihood model -LG Matrix[75]. Consurf results are displayed with Pymol software[67].

**Reporting summary.** Further information on research design is available in the Nature Research Reporting Summary linked to this article.

## Data availability

Data supporting the findings of this manuscript are available from the corresponding author upon reasonable request. A reporting summary for this Article is available as a Supplementary Information file. The source data underlying Figs. 1e–g, 2, 3a, c–f, 4, 5, 6b, d, 7a, b, d, 8c–f and Supplementary Figs. 2, 3b, c, 4, 5, 6b, c, 8, 9c, d are provided as a Source Data file. The raw data that support the findings of this study are available in FigShare with the identifier https://doi.org/10.6084/m9.figshare.8217824.

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

## Acknowledgements

We would like to thank J-M D'Ambrosío for help with plasmid construction, AC Gavin and C Beh for yeast strains and plasmids, and the ImagoSeine facility at the Institut Jacques Monod, member of IBiSA and the France-BioImaging (ANR-10-INBS-04) infrastructure. This work was supported by the CNRS, by the Agence Nationale de la Recherche Grant (ANR-16-CE13-0006), GENCI (DARI A0010710136 and A0030710136) and Marie Curie CIG grant (631997) to AC. NFL was supported by a fellowship from the Ministère de l'Enseignement Supérieur, de la Recherche et de l' Innovation.

## Author contributions

G.D. designed and supervised research. N.-F.L., M.M., and G.D. carried out site-directed mutagenesis, produced, purified and labeled all the recombinant proteins of this study. N.-F.L. and G.D. performed the in vitro experiments. R.G. performed MD simulations as well as electrostatic potential calculations. M.R. contributed tools for the analysis of MD simulations. A.C. and V.A. designed and performed the cell biology experiments. N.-F.L. performed sequence analysis. G.D., N.-F.L., R.G., A.C. and V.A. analyzed the data. A.C. participated in the refinement of the project and assisted with the manuscript. G.D. wrote the manuscript. All of the authors discussed the results and commented on the manuscript.

## Additional information

**Competing interests:** The authors declare no competing interests.

