## [Peer Review File · Nature Communications]

Reviewers' Comments:

Reviewer #1:

Remarks to the Author:

This manuscript to Nature Communications focuses on how the lid of Osh6p effects the attraction to the membrane and function. This is a well thought out study that is worthy of publication in a high-impact journal. However, there are some suggestions for improvement listed below.

General Comments:

1. Liposomes: The methods section suggest that the liposomes should be no larger than 200nm due to the filters used. However, there appears to be no quantitative measures of the liposome size via DLS or another technique. As the authors are aware, these proteins have ALPS motifs and thus membrane curvature will influence results, so quantifying liposome size might be important to show.
2. Liposomes pelleting: Past work from Schultz et al. (ref. 42) have shown that with Osh6p at 50% PS that >75% of liposomes pellet. However, studies here with 30% PS (Figure 1e) show significantly less with the wildtype. Moreover, the percent bound as a function of PS concentration is small (Figure 1f). As the simulations show, clearly the beta-loops with their positively-charged residues strongly bind to membranes. The presence of PI4P in Figure 1e appears to be the cause for this. Was PI4P also included in the Figure 1f? If not, this seems contrary to past work.
3. Simulations with potential lipid extraction motif: The authors claim that Figures S8A and S8B suggest a mode conducive for lipid extraction. However, the image and distances suggest otherwise or need more quantification. The distance metric used is not clear what quantifies the 'center of the binding pocket'. Is this the mouth opening or the inner pocket? This is closer than other simulations but doesn't appear to be deeply embedded into the membrane for lipid extraction. More details should be provided.
4. Osh6p(Delta)35 simulations without lid: Snapshots of these simulations should be provided in Figure 6 besides just saying the results are similar to Delta69. Moreover, the statement that the lid (when it closes) weakens the affinity to the membrane is not supported by the simulations. Yes the mouth region attraction is greatly reduced but then the beta-loops increase their interaction and are stabilized. So there is no indication in the tendency for Osh6p to unbind.

Specific Comment:

Molecular Dynamics: The authors consistently removed the 's' in this word throughout the manuscript, but the including the 's' is needed. Please fix.

Reviewer #2:

Remarks to the Author:

The manuscript by Lipp et al (hereafter the authors) demonstrated that the PS/PI4P-exchangeable LTP Osh6p alters its affinity for phospholipid membranes, depending on whether it encloses lipid ligand molecule. Interestingly, the lipid-bound conformation with a closed lid is shown to have lower affinity for PS-containing membranes than for neutral membranes. The experiments were well designed (except for a few points commented below) by using multi-disciplinary methods, the data presented are sound, and the manuscript has the potential to provide a novel and important mechanical insight in LTP biology. However, the current manuscript lacks several essential experiments and explanations/discussion to conclude that an electrostatic switching controls the activity of the PS/PI4P exchanger Osh6p. The most critical concern is that the main conclusion of the manuscript might be discrepant to the second law in thermodynamics. Therefore, the manuscript should be reconsidered after making appropriate revisions. Specific comments are

listed below.

Major comments:

1. The authors concluded that the open-lid-conformation is important for Osh6p to bind anionic (negatively-charged) membranes. It is an excellent idea to have employed diphytanoyl-PS. However, the lack of experiments using non-PS/PI4P anionic lipids (such as PI and PA) as the matrix anionic lipids (Fig. 4) raised the possibility that the open-lid-conformation of Osh6p binds to serine/inositol-4P groups (which is the head groups of lipid ligands for Osh6p) of membrane lipids, not simply their negative charge. The author should experimentally eliminate this possibility.

2. P3, L2: By citing a previous paper by Fairn et al, the authors depicted that the ER is almost neutral in the introduction section (P3, L1-2), while the authors argued that the ER is (at least) weakly anionic (e.g., P8, P13) in the results and discussion sections. For the main conclusion of the manuscript, it is essential to clarify whether the cytosolic surface of the ER is anionic enough to support the model of the authors (see below). Thus, the authors should present compelling evidence to clarify this point. To my opinion, the paper by Fairn et al demonstrated different distribution of PS among different organelles by using a PS-binding probe, but not analyzed other major anionic lipid types (such as PA and PI), thus giving no claim that the ER is almost neutral. I consider the authors miscited the paper of Fairn et al in the authors' context.

The major comment #3-6 are mutually related:

3. The main conclusion of this manuscript argued that Osh6p in complex with its lipid ligands (PS or PI4P) has very low affinity for anionic membranes. It may well explain a mechanism underlying how lipid-enclosed Osh6p is rapidly detached from the donor membrane. But, this model raises the question of another side: how can the lipid-enclosed Osh6p bind to acceptor membrane to release the lipid? From a thermodynamic aspect, I have the concern that the advantage in the detach step should be compensated by the disadvantage in the binding step, making no net increase in the overall exchange cycle rate. One possible explanation may be that lipid-enclosed Osh6p can bind to neutral membrane. But, this explanation is incompatible with the main conclusion (P13, L11-4up). The authors should eliminate this concern.

4. The net charge of PS is minus 1 while that of PI4P is minus 2. Thus, in the in vitro assay system, one cycle of PS/PI4P exchange transfers one minus charge from PI4P-donor liposomes to PS-donor liposomes. Nevertheless, PS/PI4P exchange was shown to efficiently proceed (Fig. 4). This looks enigmatic if the proposed electrostatic switching is really the key to control the PS/PI4P-exchange rate, considering that lipid-enclosed Osh6p should facilitate the detaching step (to extract a lipid from donor) but in turn inhibit the attaching step (to release the lipid to acceptor).

5. P12-14, MD study and Movies: The MS video data are supporting that the lid-closed form with a lipid ligand less interacted with membranes than the lid-open form. In this MD study, A lipid molecule is illustrated in the Movie 1 and 2, but this lipid molecule is not transferred to the phospholipid bilayers in the analyzed time scale main (500 ns). Why does not the MD analysis recapitulate the event for Osh6p to release the enclosed lipid to acceptor membrane? Do the MD data support my concern described above? Namely, the advantage in the detach step is compensated by the disadvantage in the binding step, making no net increase in the overall exchange rate.

6. A recent review article has proposed that a key advantage for the occurrence of LTP-demixed lipid transfer at organelle membrane contact sites is the accuracy, not the rapidity, in transport (Hanada 2018: J Lipid Res, 59, 1342). I feel that findings in the manuscript can be interpreted from the aspect of the accuracy of transport: for instance, the low affinity of PI4P-enclosed Osh6p for anionic membranes may suppress a possible misdelivery of PI4P to non-ER organelles if the surface of the ER is less anionic (negatively-charged) than other organelles. If necessary, the authors may reconsider the results from such different aspect. When the findings are not relevant

to the advantage in accurate delivery, the paradox of why Osh6p can exchange PS/PI4P between the ER and PM, in spite of co-existence of other organelles having PS and PI4P (e.g., Golgi complex, and recycling endosomes), should be explained (for this, see also comment #7).

Minor comments:

7. P3, L4up-P4, L2: How can Osh6p (an ORD alone LTP) be recruited to the patchy regions (presumably representing the ER-PM contact sites)? Because Osh6p has no clear modules to associate with the ER and PM simultaneously, Osh6p with its open-lid is expected to be distributed on the PM or ER in a uniform pattern, not in a patchy pattern. A recent paper (Venditti et al 2019: JCB, 218, 783) may give a hint to the authors to discuss a possible mechanism: e.g., Osh6p might be recruited to the contact sites by associating with an adaptor protein(s) that has a module(s) to bind to the ER and/or PM.

8. A recent study (Wang et al 2019: Mol Cell, 73, 458) showed that PI(4,5)P₂-dependent tetrameric formation of human ORP2 is required for its PI(4,5)P₂-transfer activity. Does Osh6p alter in its oligomeric state, depending on lipid ligands? This is not a question obligatory to answer. I understand the paper by Wang et al presumably appeared during the preparation or submission of this manuscript. So, if the authors would like to do so, they may comment on the question for readers.

9. P2, OSBP-related domain: OSBP is the abbreviation of oxysterol binding protein. It should be defined at the first appearance.

10. In the main text, capital letters are used to specify each panel of figures (e.g., Fig. 1A), while small letters are in the actual figures (e.g., Figure 1a).

Reviewer #3:

Remarks to the Author:

The question addressed in this paper is how does the lipid transport protein Osh6 mediate lipid transfer between ER/PM interface? Their study suggests that the closure of the 'lid' that covers the binding pocket allows for the dissociation from the membrane, similar to the results previously shown for PITPa, another lipid transporter. Whilst the experiments are convincing, there are a couple of discrepancies that need clarification.

Comments

[1] Figure 1d – Osh6p binding to liposomes in the absence of PS is greater compared to its presence. Is this significant?

[2] The Osh proteins (WT and mutant) used here have been made in E.Coli which do not contain PS or PI4P. Are they loaded with bacterial lipids such as phosphatidic acid or phosphatidylglycerol? Are the recombinant proteins in the 'open' or 'closed' configuration?

[3] Fig. 2a – In this experiment, a protein is made with a cysteine residue in the binding pocket. Thus in the 'open' state the cysteine residue would get covalently modified in the presence of the maleimide CPM. Why do the wild type protein (S190C) not get modified when PS or PI4P is present? Surely, if the protein is undergoing 'open' and 'closed' configuration on the liposomes, it would be expected that some protein would get modified despite the presence of PS or PI4P.

[4] Does the S190C protein in the absence of liposomes get modified with CPM?

[5] Fig 2e – In this panel, Osh6p binds to the liposomes more avidly when 30% diphyt-PS is present. Is this because of the negative charge of the liposomes. If so, does Osh6p bind to the liposomes if phosphatidylinositol or phosphatidic acid is also present to provide the negative charge?

[6] Fig. 7f – the truncation mutant, Δ69 which lacks the lid is able to restore function in yeast is surprising. This mutant binds avidly to membranes (Fig. 1c, e and f) even more than the lid

mutant, L69D. This would suggest that most of the truncation mutant would be unable to take part in lipid exchange at it would be such to membranes.

[7] Does this truncation mutant, $\Delta 69$, function in the PS transport assay as well?

Point-by-point response to reviewers

Reviewer #1

This manuscript to Nature Communications focuses on how the lid of Osh6p effects the attraction to the membrane and function. This is a well thought out study that is worthy of publication in a high-impact journal. However, there are some suggestions for improvement listed below.

We thank the Reviewer #1 for his/her highly positive comments on our study. We have responded to his/her main questions through additional experiments and improvements in our MD analyses.

General Comments:

1. Liposomes: The methods section suggest that the liposomes should be no larger than 200nm due to the filters used. However, there appears to be no quantitative measures of the liposome size via DLS or another technique. As the authors are aware, these proteins have ALPS motifs and thus membrane curvature will influence results, so quantifying liposome size might be important to show.

Figure 1.

The reviewer raises an interesting point. A segment [45-69] of the Osh6p's lid might have the features of a membrane-binding amphipathic helix (high hydrophobicity $\langle H \rangle = 0.55$ and moderate hydrophobic moment $\langle \mu_H \rangle = 0.3$) although it contains a potential helix breaker (Proline 60, Figure 1). However, this segment does not display ALPS-like features like the Osh4p's lid (Drin et al., NSMB, 2007) : its putative polar face is not rich in serine and threonine and it contains charged residues. Thus we did not consider that Osh6p could respond to membrane curvature. Nevertheless we think that the suggestion of Reviewer #1 is excellent as we can more clearly ascertain that the binding of Osh6p is essentially governed by the charge surface of membranes and the presence of its lipid ligands. We now report in a **new Sup. Table 1** the mean hydrodynamic radius and polydispersity of the liposomes preparation, obtained in this study by extrusion, according to their lipid composition. Data have been obtained by dynamic light scattering using a Dynapro apparatus (**described now in the M&M section**). The liposome size does not markedly change with the lipid composition (85-110 nm). We are not in the range of radii that are typically detected by an ALPS motif (30-40nm). Overall, neutrally-charged liposomes seems slightly bigger than the liposomes enriched with anionic lipids but the apparent hydrodynamic radius is influenced by the charge surface (we reported this for instance in Drin et al., NSMB, 2007 on ALPS). Our DLS data strengthen the finding that the capacity of Osh6p WT to bind to anionic membranes is directly regulated by the presence of its lipid ligands.

2. Liposomes pelleting: Past work from Schultz et al. (ref. 42) have shown that with Osh6p at 50% PS that >75% of liposomes pellet. However, studies here with 30% PS (Figure 1e) show significantly less

with the wildtype. Moreover, the percent bound as a function of PS concentration is small (Figure 1f). As the simulations show, clearly the beta-loops with their positively-charged residues strongly bind to membranes. The presence of PI4P in Figure 1e appears to be the cause for this. Was PI4P also included in the Figure 1f? If not, this seems contrary to past work.

Thank you for raising this question. Using distinct approaches (flotation and fluorescence-based assays) we show in a reproducible manner that Osh6p weakly binds to DOPC liposome containing up to 30 mol% of POPS and devoid of PI4P (Figure 1d, 1e-panel left, Figure 1f) or doped with trace amount of PI4P (Figure 1e-panel right). The explanation is that PS and PI4P are ligands of Osh6p and the protein in its loaded state has a lower affinity for anionic membranes. We note that Osh6p can bind to liposomes with 30 mol% of anionic lipids and aggregate them (Figure 4c). This is reminiscent to the work of Schulz and co-workers. However, in contrast to their study, we only see the binding of the protein to liposomes and their aggregation in the absence of its lipid ligands (PS or PI4P). Indeed, Schulz and co-workers observed that Osh6p associates with and aggregates anionic liposomes enriched with DOPS (thus a lipid ligand) via sedimentation and pelleting assays. However, they used liposomes whose composition was quite unrealistic if the idea was to understand Osh6p's function in a cellular context. The fact is that these liposomes have very extreme features as they combine a high density of anionic lipids with a loose lipid packing. Such a combination of features does not exist in the cellular membranes: the ER has a loose lipid packing yet it is weakly anionic (<20% PS and PI) whereas the inner leaflet of the PM contains up to 45% of anionic lipids (PS+PI) but has a tight lipid packing (due to saturated glycerophospholipids, sphingolipids and sterol). Worse, as pointed by Bigay & Antonny (Dev Cell, 2012), such a combination is dangerous *in vitro* as it might artificially favor membrane-protein association and give biased results due to exacerbated electrostatic and hydrophobic interactions.

To examine this point, we performed extra flotation assays using DOPC liposomes containing increasing amount (0, 30, 50, 100%) of DOPS at the expense of DOPC to produce liposomes similar to those used by Schulz and co-workers. We see that above 30% mol DOPS, Osh6p strongly binds to those liposomes and that almost 100% of protein associates with pure 100% DOPS liposomes (Figure IIA). We repeated the experiments with DOPC liposomes containing increasing mol% of POPS (the lipid ligand used in our study). Because POPS (C16:0-C18:1) has only one unsaturated lipid-chain compared to DOPS (diC18:1), the lipid packing and the density of negatively-charged lipids increases jointly. We replicated our results using liposomes with 30% POPS (as in Figure 1 and 2 in the manuscript) and measured that only 40% of Osh6p was bound to pure POPS membrane.

Furthermore, by performing DLS measurement, we observed that Osh6p was able to induce a slight aggregation of liposomes containing 50 mol% and a much more massive aggregation with pure 100% DOPS liposomes (Figure IIB). Thus, we can replicate the observations made by Schulz and coworkers but only with membranes that are quite unusual in term of lipid composition, combining a high density of anionic lipids and a loose lipid packing. Osh6p is likely anchored on such membranes in a lipid loaded state.

In the Discussion part, we added a sentence to suggest why Schulz et al observed the membrane-binding and aggregation capacity of Osh6p using very unusual lipid composition.

Figure II. **(A)** Flotation assays. *Osh6p* (750 nM) was incubated with pure DOPC liposomes (750 μ M lipids) or DOPC liposomes containing various amount of DOPS or POPS (30, 50 or 100 mol%). Data are represented as mean \pm s.e.m. ($n=3$). **(B)** Aggregation assays in real time. Liposomes (100 μ M total lipids) were mixed with *Osh6p* (500 nM final concentration). The aggregation of liposomes was followed by dynamic light scattering by measuring the mean radius of particles. Shaded area: polydispersity.

3. Simulations with potential lipid extraction motif: The authors claim that Figures S8A and S8B suggest a mode conducive for lipid extraction. However, the image and distances suggest otherwise or need more quantification. The distance metric used is not clear what quantifies the ‘center of the binding pocket’. Is this the mouth opening or the inner pocket? This is closer than other simulations but doesn’t appear to be deeply embedded into the membrane for lipid extraction. More details should be provided.

We are sorry if the Figures S8A and S8B and the Result part lacked clarity. We calculated the center of mass of the pocket by taking into account the residues within a distance not more than 5Å from the encapsulated ligand. However, we agree that this value alone, without any tridimensional representation of *Osh6p*’s pocket and of its entry was not the best way to show the proximity between the pocket and the membrane plane during the MD trajectories. **In the revised version, we now calculate the height (h_{entry}) between the entry of the pocket relative to the membrane plane (new Sup Fig. 9c).** In a new Sup Fig 9a,b, we also show

- the docking geometry of *Osh6p* Δ 35 at 300 ns and at the end of trajectory Δ 35-1
- the docking geometry of *Osh6p* Δ 69 at the end of Δ 69-2 and Δ 69-4 trajectories

with additional information: **the contour of the lipid-binding pocket** and **the position of the pocket entry**. One can now more easily appreciate, notably for the $\Delta 69-4$ trajectory, that the entry of the pocket tends to be closer to or is at the level of the water/membrane interface, when Osh6p is devoid of its lid. **The main text has been corrected accordingly.**

4. Osh6p(Delta)35 simulations without lid: Snapshots of these simulations should be provided in Figure 6 besides just saying the results are similar to Delta69.

We agree. We have added snapshots of the simulation showing the reorientation of Osh6p $\Delta 35$ on the membrane once the lid is removed from its structure. We have also extended the length of the simulation with additional 100 ns to confirm that the protein tends to adopt a membrane docking geometry similar to that of Osh6p $\Delta 69$ at the end of the $\Delta 69-1$ to $\Delta 69-3$ trajectories (**new Figure 6c and 6d**).

Moreover, the statement that the lid (when it closes) weakens the affinity to the membrane is not supported by the simulations. Yes the mouth region attraction is greatly reduced but then the beta-loops increase their interaction and are stabilized. So there is no indication in the tendency for Osh6p to unbind.

We respectfully disagree with the reviewer on this point. Several observations suggest that the presence of the lid lowers the avidity of Osh6p $\Delta 35$ for membrane comparatively to Osh6p $\Delta 69$, in line with our experimental data.

1) in the MD simulations, Osh6p $\Delta 35$ associates with the bilayer via two loops - $\beta 14-\beta 15$ and $\beta 17-\beta 18$ – whereas Osh6p $\Delta 69$ relies on these two same loops plus additional anchoring points i.e. other loops and structural elements (Figure 5d) that are otherwise masked by the lid or whose interaction with membrane is prevented due to the repulsion between the lid and the membrane.

2) Concomitantly, the interaction energies between Osh6p $\Delta 35$ and the membrane are half weaker than those calculated with Osh6p $\Delta 69$. The anionic nature of the lid and the fact that key membrane-interacting loops are masked explain this difference.

3) There is no evidence to suggest that the presence of the lid and the occlusion of the entry of the pocket can be compensated by an increased association of $\beta 14-\beta 15$ and $\beta 17-\beta 18$ loops with the membrane. On the contrary, as reported below (Figure III), the Coulomb interaction energy between these two loops and the membrane, calculated for the simulations during which Osh6p $\Delta 35$ or Osh6p $\Delta 69$ docks onto the membrane, are similar. The same is true when considering VDW interactions (not shown).

Figure III. Coulomb energy between the $\beta 14-\beta 15$ and $\beta 17-\beta 18$ loops of Osh6p and the DOPC/POPS membrane

4) there is a tendency of Osh6 Δ 35 to dissociate compared to Osh6p Δ 69. During the Δ 35-1 trajectory, the protein binds through the β 14- β 15 and β 17- β 18 loops to the bilayer during a short time (244 to 364 ns) but then the β 14- β 15 loop unbinds and, at the end of the trajectory, Osh6p Δ 35 is solely anchored to the membrane via the β 17- β 18 loop (see Supp Movie 1, also **new Sup Fig S9a and new statement in the text**). This is reflected by the distance (height) between the mass center of the protein and the membrane which increases in the 364-500 ns window of the trajectory (Figure 5b and Figure IV). Comparatively, Osh6p Δ 69 once bound to the membrane remains tightly attached (Figure 5b and Supp Figure 8b, supp Movies).

Figure IV. Evolution of the height of the center of mass of Osh6p relative to the membrane plane and that of the energy of interaction between the protein and the membrane (Coulomb and VDW) during the Δ 35-1 trajectory.

Specific Comment:

Molecular Dynamics: The authors consistently removed the 's' in this word throughout the manuscript, but the including the 's' is needed. Please fix.

This is now corrected.

Reviewer #2 (Remarks to the Author):

The manuscript by Lipp et al (hereafter the authors) demonstrated that the PS/PI4P-exchangeable LTP Osh6p alters its affinity for phospholipid membranes, depending on whether it encloses lipid ligand molecule. Interestingly, the lipid-bound conformation with a closed lid is shown to have lower affinity for PS-containing membranes than for neutral membranes. The experiments were well designed (except for a few points commented below) by using multi-disciplinary methods, the data presented are sound, and the manuscript has the potential to provide a novel and important mechanical insight in LTP biology.

However, the current manuscript lacks several essential experiments and explanations/discussion to conclude that an electrostatic switching controls the activity of the PS/PI4P exchanger Osh6p. The most critical concern is that the main conclusion of the manuscript might be discrepant to the second

law in thermodynamics. Therefore, the manuscript should be reconsidered after making appropriate revisions. Specific comments are listed below.

We thank the Reviewer #2 for considering that our work is novel and important and for acknowledging the quality of our experiments and of our multidisciplinary approaches. As detailed below, we have addressed most of his/her points through additional experiments and textual clarifications.

Major comments:

1. The authors concluded that the open-lid-conformation is important for Osh6p to bind anionic (negatively-charged) membranes. It is an excellent idea to have employed diphytanoyl-PS. However, the lack of experiments using non-PS/PI4P anionic lipids (such as PI and PA) as the matrix anionic lipids (Fig. 4) raised the possibility that the open-lid-conformation of Osh6p binds to serine/inositol-4P groups (which is the head groups of lipid ligands for Osh6p) of membrane lipids, not simply their negative charge. The author should experimentally eliminate this possibility.

We consider that PS is possibly the best lipid to build a membrane with a high density of negatively-charged lipids that mimics the inner leaflet of the PM without introducing extractable PS but we agree that PI or PA as a matrix anionic lipid can also be informative. We agree that it can reinforce the idea that the binding of the open form of Osh6p to anionic membranes is driven by non-specific electrostatic interactions. Accordingly, we included three new data sets (new Sup Fig 4a, c&d) showing that :

1) Osh6p binds to liposomes enriched with 30 mol% POPA or PI as efficiently as to those enriched with 30 mol% diphytanoyl-PS. In contrast, Osh6p weakly associates with liposomes containing 30 mol% POPS, as previously observed.

2) PI4P extraction assays show that PA and PI, like diphytanoyl-PS, are not lipid ligands of Osh6p. This means that Osh6p binds to PA and PI-rich liposomes because these liposomes are both anionic and devoid of extractable lipid ligands.

3) Osh6p WT as well as its $\Delta 35$ counterpart poorly associate with PA and PI-rich liposomes if these liposomes contain extractable lipid ligands (i.e., 5 mol% POPS or diC16:0-PI4P).

Thus, we found that Osh6p strongly binds to anionic membranes regardless of the nature of the polar head of anionic lipids if these membranes are devoid of its ligands; it binds much more weakly to these membranes only in the presence of a ligand. Summarizing, the switching mechanism is valid whatever the nature of the matrix anionic lipids: this conclusion is also important for answering the next question of Reviewer #2

2. P3, L2: By citing a previous paper by Fairn et al, the authors depicted that the ER is almost neutral in the introduction section (P3, L1-2), while the authors argued that the ER is (at least) weakly anionic (e.g., P8, P13) in the results and discussion sections. For the main conclusion of the manuscript, it is essential to clarify whether the cytosolic surface of the ER is anionic enough to support the model of the authors (see below). Thus, the authors should present compelling evidence to clarify this point. To my opinion, the paper by Fairn et al demonstrated different distribution of PS among different

organelles by using a PS-binding probe, but not analyzed other major anionic lipid types (such as PA and PI), thus giving no claim that the ER is almost neutral. I consider the authors miscited the paper of Fairn et al in the authors' context.

We thank the reviewer for raising this point. The study by Fairn et al focuses on the topology of PS in cell membranes, reporting that PS is mostly in the luminal leaflet of the ER. This paper, along with others, strongly suggests that the difference in concentration of PS in the cytosolic leaflet of the ER and PM is primarily responsible for the gradient of net anionic charge between these two membrane surfaces (as measured using charge sensors in yeast and human cell (Yeung et al., Science 2008, Haupt & Minc, MBoC, 2016)). In the first instance, to mimic this gradient of charge, we used weakly anionic ER-like liposomes containing 5% PS and PM-like liposome with 30 % of anionic lipids (25% diphytanoyl-PS + 5% PI4P). However it was measured by G. Daum and co-workers (Zinser et al., 1991) and since then, reported in many reviews (cited in this manuscript) that PI is equally distributed in the ER and the PM leaflets and account for 15 mol% of lipids in these two membranes ; PA is present in trace amount (1-4%). So the ER-membrane is weakly anionic compared to the PM but indeed cannot be considered as almost neutral. **We have corrected the Introduction part (p3) in the revised manuscript accordingly.**

However, we would like to also emphasize the following points:

- 1) Osh6p transports efficiently lipids between weakly anionic ER-like and PM-like membranes (i.e., with 5 mol% PS or PI4P, Figure 4a,b) and that the switching mechanism restricts the retention of Osh6p on more anionic membranes (i.e. 30 % anionic lipids), thereby maintaining the activity of this LTP.
- 2) Considering that the cytosolic side of the ER contains not only PS but also 15% PI, we propose that this mechanism permit Osh6p to limit its retention at the PM but also on the ER membrane.
- 3) We now report that Osh6p weakly binds to membranes enriched with PI or PA in the presence of PS or PI4P.

Thus even without knowing the precise mol% concentration and the nature of anionic lipids in the cytosolic leaflet of the ER and the PM, we can conclude that the switching mechanism allows Osh6p to work efficiently at the ER/PM interface. To further strengthen our model, we now show, **in Figure 7, new PI4P transport assays between complex ER-like membranes (PC/PI/PA/PS 76/15/4/5 mol/mol) and PM-like membranes (PC/ diphytanoyl-PS/PI/PA/PI4P 56/25/10/4/5).** We showed that Osh6pWT is as efficient in that system as in our previous experiments; in contrast, the activity of 4DA and 5A2G mutants is strongly reduced or abolished, respectively

The L69D and Δ 69 mutants, in which the lid is absent or non-functional, can be observed at the ER in yeast; we suggested that this was due to their higher avidity for anionic membrane AND because PS might accumulate to some extent at the ER due to the inability of these mutants to transport it. However, we should emphasize that the cellular localization of these mutants is in fact very complex: in some cells, strong ER localization pattern can be observed, which is easily recognizable and underscored by colocalization with the Sec63 ER marker (Fig. S1). However, our images also show the binding of these mutants to other membranes. Note the numerous puncti in many cells, whose nature cannot be determined, but could represent various cellular compartments and/or aggregation

of the mutant proteins, some enrichment on vacuoles, and, importantly, quite strong fluorescent signal at the cell surface. It is virtually impossible to distinguish between cortical ER and plasma membrane localization by fluorescent microscopy, and therefore it is difficult to determine to what extent the cell surface signal represents one or the other compartment. Thus, reconsidering the fact that intracellular membranes, due to the presence of PI, PA and PS have at least 20% of negatively-charged lipids, we now judge better to essentially highlight in the Discussion that these two mutants bind to internal membranes due to their higher avidity for negatively-charged lipids present at their surface. **We have rewritten a part of the Discussion to emphasize more the complex pattern of the cellular localization of these mutants.**

The major comment #3-6 are mutually related:

3. The main conclusion of this manuscript argued that Osh6p in complex with its lipid ligands (PS or PI4P) has very low affinity for anionic membranes. It may well explain a mechanism underlying how lipid-enclosed Osh6p is rapidly detached from the donor membrane. But, this model raises the question of another side: how can the lipid-enclosed Osh6p bind to acceptor membrane to release the lipid? From a thermodynamic aspect, I have the concern that the advantage in the detach step should be compensated by the disadvantage in the binding step, making no net increase in the overall exchange cycle rate. One possible explanation may be that lipid-enclosed Osh6p can bind to neutral membrane. But, this explanation is incompatible with the main conclusion (P13, L11-4up). The authors should eliminate this concern.

We appreciate this comment. Osh6p has a low affinity for neutral membrane and a low affinity for anionic membranes yet only when it is in the presence of its ligands as stated by the Reviewer. We agree that this weak affinity might result from the combination of a higher dissociation rate constant, due to the closing of the lid, and a slower rate of association of Osh6p to the membrane because the lid is anionic and must open. Nevertheless, our main point is that this weak affinity explains the fact that Osh6p remains mostly soluble over time during transport cycles (Figure 4c). Accordingly, we measured and indicated that Osh6p preserves a high transport capacity between highly anionic membranes but this activity is lower than that measured with more neutral membranes (Figure 4a,b). **We have neither measured nor stated, contrary to the Reviewer's suggestion, that the switching mechanism allows a net increase in the overall exchange rate.** We only concluded that the switching mechanism allows Osh6p to keep a residency time on anionic membranes almost as low as on neutral membranes and thereby to maintain its transport efficiency in that context. Without the lid, or when the lid is not anionic, Osh6p transfer activity is abolished between anionic membranes.

Except the experiments shown in the Fig3C, in which we measured in real-time, the dissociation of Osh6p from liposomes upon adding PS, all our other measurements (flotation, fluorescence-based assays) are done at steady state. **Thus we rephrased some sentences in the manuscript that possibly suggested that the weak affinity of the loaded form of Osh6p solely relies on the dissociation step.**

4. The net charge of PS is minus 1 while that of PI4P is minus 2. Thus, in the in vitro assay system, one cycle of PS/PI4P exchange transfers one minus charge from PI4P-donor liposomes to PS-donor liposomes. Nevertheless, PS/PI4P exchange was shown to efficiently proceed (Fig. 4). This looks

enigmatic if the proposed electrostatic switching is really the key to control the PS/PI4P-exchange rate, considering that lipid-enclosed Osh6p should facilitate the detaching step (to extract a lipid from donor) but in turn inhibit the attaching step (to release the lipid to acceptor).

This is a good suggestion. Before answering, we emphasize that the major driving force of PS/PI4P exchange between two membranes is the respective concentration of accessible PS and PI4P in each membrane. In Moser von Filseck et al., 2015, we showed *in vitro* that Osh6p exchanges and equilibrates, according to the law of mass action, a PS pool initially present in ER-like membranes (at 5mol%) with a PI4P pool present in PM-like membranes (5mol%). In yeast, the maintenance of a PI4P gradient at the ER/PM interface allows Osh6p to transport PS unidirectionally.

Here, we show that a second mechanism controls Osh6p's activity, allowing Osh6p to remain weakly bound to membranes whatever the density of negatively-charged lipids (in a large range, from 0 to up to 35%, if we consider -2 for PI4P), in the presence of lipid ligands, and to be efficient. Therefore it is unlikely that a slight variation of negative charge at the surface of the ER- and PM-like membranes due to PS/PI4P exchange would impact the overall activity of Osh6p. *In vitro*, once the exchange process is completed and the lipids are equilibrated, we can presume that the mol% in term of net negative charge would be increased by 2.5 in ER-like and decreased by 2.5 in the PM-like membrane. This is low in comparison with the global negative charge provided by 25 mol% of diphytanoyl-PS.

5. P12-14, MD study and Movies: The MS video data are supporting that the lid-closed form with a lipid ligand less interacted with membranes than the lid-open form. In this MD study, A lipid molecule is illustrated in the Movie 1 and 2, but this lipid molecule is not transferred to the phospholipid bilayers in the analyzed time scale main (500 ns). Why does not the MD analysis recapitulate the event for Osh6p to release the enclosed lipid to acceptor membrane? Do the MD data support my concern described above? Namely, the advantage in the detach step is compensated by the disadvantage in the binding step, making no net increase in the overall exchange rate.

Our MD analyses as well as those previously published by expert groups in the field (Klauda's or Vattulainen's group) are very useful to predict at the atomic level how LTPs (e.g., Osh4p, PITP α) docks onto membranes. As stated by Reviewer #2, our simulations, in very good agreement with our experimental data, allows analyzing from an energetic standpoint why the open and closed-form of Osh6p differ in their ability to interact with anionic membranes. However, the length of current all-atom MD simulations with membrane/protein/water systems, (here 500 ns, or $\geq 1\mu\text{s}$ with Osh4p-ergosterol complex, Rogaski and Klauda, JMB, 2012 or PITP α in open or closed configuration, Grabon et al., JBC, 2017), is insufficient to describe how an LTP extracts or delivers a lipid molecule. Such processes, which likely occur on a millisecond scale, are not accessible due to the size of the model systems and the computational means and time that would be required for performing these lengthy simulations. The mechanism of lipid release can be potentially analyzed by pulling the ligand out of its pocket by biased MD simulations - steered-MD simulations (Osh4p in solution) or steered-MD simulations+ umbrella sampling (PITP α on membrane). Performing such analyses is beyond the scope of this study.

6. A recent review article has proposed that a key advantage for the occurrence of LTP-demixed lipid transfer at organelle membrane contact sites is the accuracy, not the rapidity, in transport (Hanada 2018: J Lipid Res, 59, 1342). I feel that findings in the manuscript can be interpreted from the aspect of the accuracy of transport: for instance, the low affinity of PI4P-enclosed Osh6p for anionic membranes may suppress a possible misdelivery of PI4P to non-ER organelles if the surface of the ER is less anionic (negatively-charged) than other organelles. If necessary, the authors may reconsider the results from such different aspect. When the findings are not relevant to the advantage in accurate delivery, the paradox of why Osh6p can exchange PS/PI4P between the ER and PM, in spite of co-existence of other organelles having PS and PI4P (e.g., Golgi complex, and recycling endosomes), should be explained (for this, see also comment #7).

Hanada's review offers relevant thinking and assumptions on why many LTPs work in contact sites. Because these LTPs are found in narrow gaps between two organelles and are often attached to them via specific targeting modules, this might guarantee the accuracy of transport processes (no lipid leak towards other organelles) rather than higher transport rates. In our manuscript we address a quite different question, how the lipid-encapsulating domain of an LTP can limit its residency time on membranes to allow efficient transport? We demonstrate that the ability of Osh6p to bind to membrane remains insensitive to the density of negatively-charged lipids when its ligands are available. This explains how this protein can cycle rapidly between anionic membranes. But this also means that Osh6p cannot discriminate between the ER-membrane, the PM and the membrane of other organelles solely on the basis of the density in negatively-charged lipids. The channeling/accuracy of PS and PI4P exchange at the ER/PM interface is likely given by the localization of the lipid sources and their turn-over (PS in the ER, PI4P in the PM). It is not clear yet whether Osh6p delivers PS or not to the Golgi apparatus in which PI4P is also produced. Protein/protein interactions are likely partially in play to ensure the accuracy of PS/PI4P exchange, yet their role remains to be clarified (see below).

A short paragraph has been now included in the Discussion part to address this point.

Minor comments:

7. P3, L4up-P4, L2: How can Osh6p (an ORD alone LTP) be recruited to the patchy regions (presumably representing the ER-PM contact sites)? Because Osh6p has no clear modules to associate with the ER and PM simultaneously, Osh6p with its open-lid is expected to be distributed on the PM or ER in an uniform pattern, not in a patchy pattern. A recent paper (Venditti et al 2019: JCB, 218, 783) may give a hint to the authors to discuss a possible mechanism: e.g., Osh6p might be recruited to the contact sites by associating with an adaptor protein(s) that has a module(s) to bind to the ER and/or PM.

We are in full agreement with the reviewer that the results presented in this study, which address the interaction of Osh6p with the lipid component of cellular membranes, do not explain the partial and patchy distribution of Osh6p at the cellular cortex, which has been well documented. Protein-protein interactions are an important means of localizing proteins in cells, and we have in fact already identified one binding partner of Osh6p that is important for its cellular localization (A Copic and V Albanese, unpublished data). However, this work is beyond the scope of this study and will be

presented elsewhere. **We have added some discussion of the role of protein-protein interactions to the Discussion.**

We hypothesize that in yeast cells, where its lipid ligands are available, Osh6p WT is rarely in an open state, so it is largely impossible to isolate or observe it in this state. Accordingly, the WT protein is largely cytosolic or confined to contact sites, but it cannot be observed localizing to the PM or the ER in a uniform manner (Fig1a, b,c). In contrast, the mutants $\Delta 69$ and L69D, which we demonstrate to correspond to an open state of Osh6p, are not observed in patches at the cell cortex and dispersed in the cytosol, but distributed on different membranes in a rather uniform pattern, suggesting that mutations change their membrane-binding capacity. This is what is seen in Fig1c and FigS1 when these proteins are expressed tagged to mCherry in $\Delta osh6\Delta osh7$ yeast.

8. A recent study (Wang et al 2019: Mol Cell, 73, 458) showed that PI(4,5)P₂-dependent tetrameric formation of human ORP2 is required for its PI(4,5)P₂-transfer activity. Does Osh6p alter in its oligomeric state, depending on lipid ligands? This is not a question obligatory to answer. I understand the paper by Wang et al presumably appeared during the preparation or submission of this manuscript. So, if the authors would like to do so, they may comment on the question for readers.

We have no indication of a capacity of Osh6p to adopt a tetrameric form when loaded with one of its ligand. During purification by size exclusion chromatography of Osh6p loaded with PI4P, PS or in an empty state, we observe no differences in their respective elution profiles (see for instance a comparison between the elution profiles of the Osh6pWT preloaded or not with PI4P, Figure V). In the study of Wang et al, the tetramerization of ORP2 notably arises from intermolecular interactions between the monomers mediated by the lid region (the lid being in a partially open state) as well as by PIP₂. No similar configuration of the lid is seen in the crystal structure of Osh6p in complex with PS or PI4P and these lipids do not promote any interaction between monomers.

Figure V. Elution of Osh6p and Osh6p-PI4P complex using a Sephacryl S200 HR XK16-50 column. The protein corresponds to the major peak at 80 mL

9. P₂, OSBP-related domain: OSBP is the abbreviation of oxysterol binding protein. It should be defined at the first appearance.

This is now corrected

10. In the main text, capital letters are used to specify each panel of figures (e.g., Fig. 1A), while small letters are in the actual figures (e.g., Figure 1a).

Thank you. This is now corrected.

Reviewer #3 (Remarks to the Author):

The question addressed in this paper is how does the lipid transport protein Osh6 mediate lipid transfer between ER/PM interface ? Their study suggests that the closure of the 'lid' that covers the binding pocket allows for the dissociation from the membrane, similar to the results previously shown for PITPa, another lipid transporter. Whilst the experiments are convincing, there are a couple of discrepancies that need clarification.

We are grateful to Reviewer #3 for considering that our manuscript are convincing.

Comments

[1] Figure 1d – Osh6p binding to liposomes in the absence of PS is greater compared to its presence. Is this significant ?

In this first set of experiment we saw a slightly higher binding to PC liposomes than to liposomes with 5 mol% of PS or more (Figure1d) . However, overall, in many other experiments, when comparing the binding of Osh6p to DOPC liposome and to DOPC/POPS 7/3 liposomes (Figure 2e), we did not measure real differences. Similar results are obtained in fluorescence-based experiments and also with Osh6p Δ 35. Therefore we consider this binding is not very significant and shows that Osh6p^{WT} in complex with its ligands poorly binds to membranes with a density of anionic lipids ranging from 0 to 30%. We repeated a few extra experiments with Osh6p^{WT} along with Osh6p Δ 69 and Osh6pL69D using a range of POPS concentration (0, 5,10, 30 mol%) to strengthen this point and improve the Fig1d and 1f.

[2] The Osh proteins (WT and mutant) used here have been made in *E.Coli* which do not contain PS or PI4P. Are they loaded with bacterial lipids such as phosphatidic acid or phosphatidylglycerol? Are the recombinant proteins in the 'open' or 'closed' configuration ?

We analyzed (subcontracting Novalix, France) by ESI mass spectrometry one preparation of Osh6p, purified from *E.Coli*. Mass spectrum, recorded under native conditions, showed that Osh6p was partially adducted with a molecule of ~ 720 Da. The proportion of Osh6p-fortuitous ligand complexes was about 30%. LC-MS experiments were then carried out using an UHPLC system and a reverse phase LC column coupled to an ion trap mass spectrometer operating in the negative ion mode. These analyses revealed the presence of 4 major singly-charged species with mass-to-charge ratios of 719.4 m/z, 717.4 m/z, 691.4 m/z and 733.4 m/z (Figure IV.A). The first two are the most intense, which is in good agreement with the native MS data. These two species are likely 34:0 and 34:1 PE (POPE) considering lipidomic analysis of *E.Coli* (e.g., Matyash V et al., J Lipid Res. 2008 and more recently, Jeucken A et al., Cell Report, 2019)

Thus these analyses suggest that Osh6p is partially loaded with a PE molecule. However, PE is not a true ligand of Osh6p. This has been previously showed by Maeda et al. (Nature, 2013, 501(7466):257-61), who found that Osh6p isolated from yeast, where PS and PE co-exist with similar abundance, only contain PS. Thus we think that in our experiments, a little of PE is possibly released by Osh6p into the membrane, where it acts as an inert lipid like PC and does not influence our assays. Here, for the revision of the manuscript, we examined whether PA or PI could prevent Osh6p from

extracting PI4P (**new Sup Fig.4**) but also the inhibitory action of PE (Figure IV B). We measured that an excess of PE (10 mol%) in the membrane, unlike PS, does not block PI4P extraction by Osh6p, meaning that Osh6p has no avidity for PE. We thus confirm that PE is not a ligand of Osh6p.

Figure VI. LC-MS analyses of Osh6p. (A) Base peak chromatograms. Colored traces correspond to extracted ion chromatograms of the major species whose mass spectra are displayed on the right. (B) PI4P extraction assay. NBD-PH_{FAPP} (250 nM) was incubated with DOPC liposomes (80 μM lipids) containing 2% diC16-PI4P or, additionally, 10% of POPS or POPE. The fluorescence spectra of the probe were measured (λ_{ex} = 460 nm) before and after adding Osh6p (3 μM). Control spectra were recorded with liposomes devoid of PI4P to normalize the signal. The percentage of extracted PI4P was calculated by normalizing the fluorescence emission at 536 nm. Data are represented as mean \pm s.e.m (n=4).

[3] Fig. 2a – In this experiment, a protein is made with a cysteine residue in the binding pocket. Thus in the ‘open’ state the cysteine residue would get covalently modified in the presence of the maleimide CPM). Why do the wild type protein (S190C) not get modified when PS or PI4P is present ? Surely, if the protein is undergoing ‘open’ and ‘closed’ configuration on the liposomes, it would be expected that some protein would get modified despite the presence of PS or PI4P.

In the presence of PS or PI4P, we see a slow increase in CPM fluorescence signal within 30 minutes (Figure S2), which suggests that the protein is slightly modified and not completely “frozen” with one PS or PI4P molecule inside its pocket. Osh6p remains dynamic and can undergo configuration changes but it should be most of the time in a closed state. **This is now indicated in the Result part.** This is in line with the data in Figure4c reporting that Osh6p is mostly soluble at steady state in the presence of membranes containing a small amount of PS and PI4P (5%), i.e. in conditions close to those of CPM assays (with 2 mol% PS or PI4P).

[4] Does the S190C protein in the absence of liposomes get modified with CPM ?

We tested this condition and determined that Osh6p in solution was modified by CPM, suggesting that Osh6p (WT and L69D) molecules can frequently adopt an open state in solution. **These data are now included in a revised version of Figure 2 and Supp Figure 2 and commented in the Results part.** We noted that the CPM signal is higher in the presence of DOPC liposomes than in buffer alone; this suggests that the slight binding of Osh6p to pure DOPC liposomes tends to promote an open state of the protein and thus the accessibility of C190 to CPM. For Osh6p WT S190C one can also suggest that the protein is partially loaded with PE (30% of protein, see previous response) and closed, but that it fully open in the presence of DOPC liposomes, delivering PE to these liposomes.

[5] Fig 2e – In this panel, Osh6p binds to the liposomes more avidly when 30% diphyt-PS is present. Is this because of the negative charge of the liposomes. If so, does Osh6p bind to the liposomes if phosphatidylinositol or phosphatidic acid is also present to provide the negative charge ?

As indicated previously, we now show in **new Supplementary Fig4a, b & d** that Osh6p can strongly bind to liposomes enriched with 30 % PA or PI but loses its avidity for such membranes if they contain POPS or PI4P. Thus Osh6p binds, via non-specific electrostatics interactions, to liposomes enriched with diphytanoyl-PS, PA or PI because of their negative charge and because they do not include extractable ligands.

[6] Fig. 7f – the truncation mutant, $\Delta 69$ which lacks the lid is able to restore function in yeast is surprising. This mutant binds avidly to membranes (Fig. 1c, e and f) even more than the lid mutant, L69D. This would suggest that most of the truncation mutant would be unable to take part in lipid exchange at it would be such to membranes.

This is a good point. We refer to the study of Kozminsky and co-workers (J. Cell Science, 2017) who suggest that Osh6p can substitute for Osh4p, restoring polarized exocytosis and thus growth in Δ osh yeast, by extracting and transporting PI4P from light-density secretory vesicles. According to this study and the study by Ling et al., MBoC 2014, the removal of PI4P from secretory vesicles is essential to allow the loading of Sec4p to produce docking-competent vesicles. Osh6p L69D and HH/AA failed to substitute for Osh4 (our study and Kozminsky's paper), which is consistent with their inability to extract and transport PI4P (this study, Moser von Filseck, Science, 2015). In contrast, the $\Delta 35$, 5A2G, and $\Delta 69$ mutants with a full or residual ability to transport PI4P (Figure 4a,b and 7), restore growth like Osh6p WT. This suggests to us that even a minimal ability of Osh6p to transport PI4P is sufficient to restore growth of Δ osh yeast. The Δ osh growth assay therefore supports our observations in vitro that Osh6p 5A2G is able to capture PI4P despite the drastic substitution of a highly-conserved negative sequence in the lid region. **We have added additional explanation of the growth assay experiment in the Results part.** It is more difficult to explain the behavior of the Osh6p $\Delta 69$ mutant in this assay. We note that in vitro, this mutant retains a residual PI4P transport activity (Moser von Filseck et al, 2015, this work), likely because the polar head of PI4P is mostly recognized by residues in the core of the protein, not in the lid. Unfortunately, we do not have a way to monitor the PI4P transport activity of Osh6p in cells to directly compare different Osh6 mutants under physiological conditions, and we find that the steady-state cellular distribution of PI4P is too strongly affected by factors other than Osh6p (our unpublished data). One hypothesis, which would be very difficult to test, is that the protein binds to light-density secretory vesicles, sequestering PI4P molecules. Testing this hypothesis is beyond the scope of this study.

[7] Does this truncation mutant, $\Delta 69$, function in the PS transport assay as well?

Osh6p $\Delta 69$ does not transport PS in vitro (Moser von Filseck et al., 2015) and in yeast (this study), because PS recognition, contrary to PI4P recognition, relies mainly on residues of the lid (Moser von Filseck et al, 2015 ; Maeda et al, 2013).

Reviewers' Comments:

Reviewer #1:

Remarks to the Author:

The revision responded to my critiques and I have not further issues with this manuscript.

Reviewer #2:

Remarks to the Author:

I studied the revised manuscript and was satisfied.

Reviewer #3:

Remarks to the Author:

The revised version of this manuscript has answered some of the questions raised. There are still a small number of issues that need clarification.

In their rebuttal letter, their response to the following comment was:

[1] Figure 1d – Osh6p binding to liposomes in the absence of PS is greater compared to its presence.

Is this significant ?

In this first set of experiment we saw a slightly higher binding to PC liposomes than to liposomes with 5 mol% of PS or more (Figure1d). However, overall, in many other experiments, when comparing the binding of Osh6p to DOPC liposome and to DOPC/POPS 7/3 liposomes (Figure 2e), we did not measure real differences. Similar results are obtained in fluorescence-based experiments and also with Osh6p Δ 35. Therefore we consider this binding is not very significant and shows that Osh6p^{WT} in complex with its ligands poorly binds to membranes with a density of anionic lipids ranging from 0 to 30%. We repeated a few extra experiments with Osh6p^{WT} along with Osh6p⁶⁹ and Osh6p^{L69D} using a range of POPS concentration (0, 5, 10, 30 mol%) to strengthen this point and improve the Fig1d and 1f.

In the original manuscript, the Fig. legend to 1d and 1f stated that the data are represented as mean \pm s.e.m. (n=3). The Figure legend is not changed in the revised manuscript although in their rebuttal letter, they emphasise they repeated a few extra experiments to strengthen the point and improve the Figs. 1d and 1f. It is noted that the actual Fig 1f has changed but not the Figure Legends. Can the Figure legends be updated if more experiments have been included.

Fig. 1d – Can a more representative blot be provided for Fig. 1d? In the current blot (same as in the original submission) there is a marked visual difference between no PS and with 5% PS. As the text says there is no difference, a blot reflecting that statement should be used. On a separate note, have they tested for linearity of their western blot signals in the range used in the experiments.

Point-by-point response

Response to Reviewer #3.

In the original manuscript, the Fig. legend to 1d and 1f stated that the data are represented as mean \pm s.e.m. (n=3). The Figure legend is not changed in the revised manuscript although in their rebuttal letter, they emphasise they repeated a few extra experiments to strengthen the point and improve the Figs. 1d and 1f. It is noted that the actual Fig 1f has changed but not the Figure Legends. Can the Figure legends be updated if more experiments have been included.

We apologize for this mistake. For Osh6p WT, n=4-7; for Osh6p Δ 69, n=5-6 ; for Osh6p(L69D), n=5-7. The n values are not exactly the same for all the data points as in addition to measurements done with a set of liposomes having increasing %mol of POPS (0, 5, 10 or 30 mol%), we have integrated a few available values coming from other independent experiments in which only one type of liposomes is used (with 0 or 5 or 30 mol% of POPS for instance). The correct n values are now included in the caption of Figure 1 for Osh6p WT and the mutants. The conclusion remains exactly the same: Osh6p Δ 69 and Osh6p(L69D) increasingly binds to liposomes if the density of PS increases whereas Osh6p WT does not.

Fig. 1d – Can a more representative blot be provided for Fig. 1d? In the current blot (same as in the original submission) there is a marked visual difference between no PS and with 5% PS. As the text says there is no difference, a blot reflecting that statement should be used. On a separate note, have they tested for linearity of their western blot signals in the range used in the experiments.

We now provide in Figure 1d a new picture of a flotation experiment, done with Osh6p WT, which is representative of several flotation experiments. We analyze our flotation assays by SDS-PAGE, with a Sypro staining, as in all of our recent papers. In those experiments, the quantity of protein in top fractions loaded onto the gel is typically between 0 and \sim 0.9 μ g (with 0.9 μ g corresponding to the quantity of the 100% total lane). We know that the signal increases linearly with the amount of protein in this range. As an example, we show below a routine gel used to quantify a batch of Osh6p. Increasing amounts of BSA are deposited (from 0 to 2 μ g/lane) and the intensity of the band after Sypro staining allows building a standard curve representing the intensity as a function of the amount of protein. It is perfectly linear and allows here to determine the quantity of Osh6p.